# Offset of MODIS land surface temperatures from in situ air temperatures in the Upper Kaskawulsh Glacier region (St. Elias mountains) indicates near-surface temperature inversions

Ingalise Kindstedt[1], Kristin M. Schild[1,2], Dominic Winski[1,2], Karl Kreutz[1,2], Luke Copland[3], Seth Campbell[1,2], and Erin McConnell[1]

[1]Climate Change Institute, University of Maine, Orono, Maine, USA
[2]School of Earth and Climate Sciences, University of Maine, Orono, Maine, USA
[3]Department of Geography, Environment and Geomatics, University of Ottawa, Ottawa, Ontario, Canada

**Correspondence:** Ingalise Kindstedt (ingalise.kindstedt@maine.edu)

**Abstract.** Remote sensing data are a crucial tool for monitoring climatological changes and glacier response in areas inaccessible for in situ measurements. The Moderate Resolution Imaging Spectroradiometer (MODIS) land surface temperature (LST) product provides temperature data for remote glaciated areas where air temperature measurements from weather stations are sparse or absent, such as the St. Elias Mountains (Yukon, Canada). However, MODIS LSTs in the St. Elias Mountains have been found in prior studies to show an offset from available weather station measurements, the source of which is unknown. Here, we show that the MODIS offset likely results from the occurrence of near-surface temperature inversions rather than from the MODIS sensor's large footprint size or from poorly constrained snow emissivity values used in LST calculations. We find that an offset in remote sensing temperatures is present not only in MODIS LST products, but also in Advanced Spaceborne Thermal Emissions Radiometer (ASTER) and Landsat temperature products, both of which have a much smaller footprint (90-120 m) than MODIS (1 km). In all three datasets, the offset was most pronounced in the winter (mean offset >8°C), and least pronounced in the spring and summer (mean offset <2°C). We also find this enhanced seasonal offset in MODIS brightness temperatures, before the incorporation of snow surface emissivity into the LST calculation. Finally, we find the MODIS LST offset to be consistent in magnitude and seasonal distribution with modeled temperature inversions, and to be most pronounced under conditions that facilitate near-surface inversions, namely low incoming solar radiation and wind speeds, at study sites Icefield Divide (60.68°N, 139.78°W, 2,603 m a.s.l) and Eclipse Icefield (60.84°N, 139.84°W, 3,017 m a.s.l.). Although these results do not preclude errors in the MODIS sensor or LST algorithm, they demonstrate that efforts to convert MODIS LSTs to an air temperature measurement should focus on understanding near-surface physical processes. In the absence of a conversion from surface to air temperature based on physical principles, we apply a statistical conversion, enabling the use of mean annual MODIS LSTs to qualitatively and quantitatively examine temperatures in the St. Elias Mountains and their relationship to melt and mass balance.

# 1 Introduction

In recent decades, the high latitudes (>60°) have warmed at a more rapid rate than the rest of the planet, with impacts extending to distant lower latitude regions (Winton, 2006; Serreze and Barry, 2011; You et al., 2021). In particular, the loss of high-latitude glaciers has reduced the Earth's albedo (which can further accelerate warming) and contributed to global sea level rise (Budyko, 1969; Lian and Cess, 1977; Serreze and Barry, 2011; Zemp et al., 2019; Hugonnet et al., 2021). The St. Elias mountains are situated on the border of Alaska and the Yukon in a region experiencing pronounced warming and glacier mass loss compared to the rest of the high latitudes (Farinotti et al., 2019; Zemp et al., 2019; Hugonnet et al., 2021). Alaskan glaciers alone have contributed over 25% of observed sea level rise to date, the largest contribution of any one glaciated region, excluding the Greenland and Antarctic Ice Sheets (Zemp et al., 2019; Hugonnet et al., 2021). Additionally, Alaskan glaciers are losing mass at some of the highest rates globally ($-66.7 \, \mathrm{Gt \, yr^{-1}}$), and therefore remain pertinent to projections of global sea level rise (Hugonnet et al., 2021). The greater North Pacific cordillera (high elevation sectors of Alaska and neighboring parts of the Yukon and British Columbia) contains over 40 mm of global sea level rise potential in a combination of large icefields and small alpine glaciers, making widespread monitoring of glacier mass in the region a worthwhile endeavor (Farinotti et al., 2019).

Because of the influence of atmospheric temperature on surface energy balance via downward longwave radiation and sensible heat transfer, glacier mass loss is largely associated with atmospheric warming (Cuffey and Paterson, 2010). In order to better predict the impacts of projected atmospheric warming, we need to monitor temperature change and glacier response. However, due to the inaccessibility of many high-latitude regions for in situ measurements, our understanding of the region's climatic behavior relies heavily on remote sensing products, such as Moderate Resolution Imaging Spectroradiometer (MODIS) land surface temperatures (LSTs). Temperatures derived using remote sensing techniques are definitionally not measured directly. Instead, they are inferred from measurements taken by satellite sensors of the energy emitted by the earth's surface. A variety of temperature products can be obtained using remote sensing methods, including the final surface temperature product, as well as "brightness temperature", or the temperature of a perfect blackbody emitter under the same conditions. In contrast, temperatures measured in situ are directly measured using instruments onsite, and can be measured for both the earth's surface and the air above it. Surface temperatures measured in situ provide important validation for remote sensing surface temperatures such as MODIS LSTs. However, because of a lack of in situ surface temperature data in our study region, unless otherwise stated, all in situ temperatures used here refer to the air 2 m above the land surface. Our study is therefore not a standard validation of the MODIS LST product, but rather an evaluation of its use in conjunction with in situ air temperatures to characterize the near-surface temperature conditions of the St. Elias region.

MODIS LSTs are a valuable tool for monitoring climate in remote regions because they provide more than two decades (2000-present) of near-daily imagery under clear-sky conditions. However, MODIS LSTs have been observed to be lower than in situ surface and air temperatures at a number of snow- and ice-covered sites. For example, at Summit, Greenland, 2008-2009 MODIS LSTs were an average of 5.5°C lower than coincident 2 m air temperatures, amounting to an ~3°C offset in the MODIS LSTs once the difference between surface and air temperatures was accounted for (Koenig and Hall, 2010).

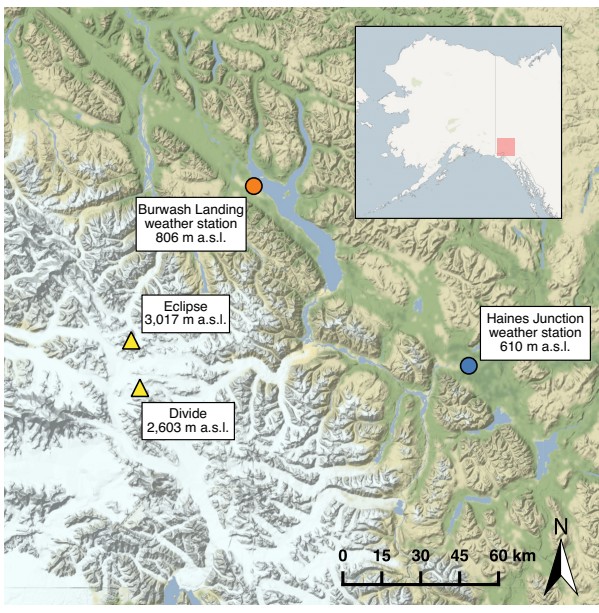

**Figure 1.** Study sites Eclipse and Divide (yellow triangles) and nearby weather station locations at Burwash Landing (orange circle) and Haines Junction (blue circle).

Likewise, in Svalbard, wintertime MODIS LSTs from a snow-covered permafrost site showed a cold offset of 1.5°C to 6°C (mean = 3°C) relative to in situ surface temperatures (Westermann et al., 2012), and MODIS LSTs from the Austfonna ice cap during 2004-2011 showed a cold offset relative to both in situ surface (RMSE = 5.3°C) and air (RMSE = 6.2°C) temperatures (Østby et al., 2014). In this study, we focus on an observed cold offset in MODIS LSTs from automated weather station (AWS) temperatures in the glaciated Upper Kaskawulsh-Donjek region of the St. Elias Mountains (Yukon, Canada; hereafter referred

to as "St. Elias"). In this region, average daily MODIS LSTs have been shown to be colder than downscaled and observed air temperatures by 5–7°C when snow cover was >90% (Williamson et al., 2017).

Remote sensing temperature products are especially useful for relating glacier behavior and mass balance to climatological changes in rugged alpine regions where glaciers tend to be at higher elevations than most nearby weather stations. Our study sites in the St. Elias are located above 2,500 m a.s.l., while nearby Environment and Climate Change Canada weather stations

are located at 610 m a.s.l. (Haines Junction) and 806 m a.s.l. (Burwash Landing; Fig. 1). Lower elevation sites are in contact with different air masses and are sensitive to different sources of variability than their high elevation counterparts, so data from these stations are not necessarily representative of climatic behavior at glaciated alpine sites (McConnell, 2019). In particular, low elevation sites are primarily sensitive to local climate, while higher elevation sites are sensitive to atmospheric circulation patterns on a large spatial scale (e.g. Alaska Range-central tropical Pacific teleconnections; Winski et al., 2018). Additionally,

low-elevation weather stations likely underestimate the warming experienced at nearby higher elevation sites. Modeling studies (Chen et al., 2003; Giorgi et al., 1997) predict that warming rates increase with elevation. Although not a universal phenomenon

(Ohmura, 2012), elevation-enhanced warming has been observed in a number of alpine mountain ranges including the St. Elias and greater North Pacific cordillera (Williamson et al., 2020; Diaz et al., 2014; Pepin et al., 2015; Rangwala and Miller, 2012).

Because we lack paired in situ surface and air temperature measurements in the St. Elias, it is not clear whether the MODIS LST offset in the region results from the instrumentation and algorithm used to produce MODIS LSTs or whether it is a real temperature difference between the air and surface. Unlike weather stations, which measure air temperature at a point typically 2 m above the surface, MODIS LSTs record the temperature of the surface itself across a 1 km$^2$ grid cell. Although air temperature and surface temperature are closely related, they are distinct and their response to the same forcing can differ (Jin and Dickinson, 2010). Cold offsets in MODIS LSTs at Summit, Greenland have been attributed to near-surface temperature inversions, which occur when the surface is colder than the air directly above it (Adolph et al., 2018). Near-surface temperature inversions develop over glaciated regions when heat transfer from the surface to the air occurs as a result of an energy imbalance at the surface-air interface (Adolph et al., 2018). Such energy imbalances can occur under low incoming solar radiation, when upward longward radiation emitted by the earth's surface may exceed downwelling energy fluxes (Adolph et al., 2018). Snow surfaces often have a high emissivity (0.949-0.997 in the 10.5-12.5 $\mu$m range; Hori et al., 2006) relative to the atmosphere, which has been observed to be as low as 0.4, depending on water vapor content (Herrero and Polo, 2012). This difference in emissivities requires the snow surface to cool relative to the air above as it equilibrates (Hudson and Brandt, 2005). One hypothesis for the offset in MODIS LSTs in the St. Elias is the presence of near-surface temperature inversions similar to those observed at Summit, Greenland. However, unlike the interiors of large ice sheets, alpine environments are characterized by heterogeneity in surface type, elevation, aspect, incline, wind scouring, and shading (note the many ridges and nunataks shown in Fig. 2), all of which affect surface energy balance. Conditions from Summit, Greenland therefore cannot be used to infer near-surface temperature inversions in the St. Elias, and to our knowledge, such inversions have not to date been observed in other alpine regions. Here, we use the term "inversion" specifically in reference to temperature inversions within 2 m of the land surface ("near-surface"). We both evaluate the plausibility of near-surface temperature inversions in the St. Elias and test two alternative hypotheses to explain the offset in MODIS LSTs in the region.

First, the LST offset could result from the large (1 x 1 km) footprint (grid cell) of the MODIS sensor. The heterogeneity of the St. Elias' environment (surface type, elevation, aspect, incline, wind scouring, shading) may not be well represented by the average temperature value of a MODIS grid cell. A second cause of the LST offset could be incorrect definition of emissivity values used to calculate MODIS LSTs from brightness temperatures. Since snow does not emit radiation uniformly, emissivity is not uniform across snow surface types, particularly in locations such as the St. Elias icefields, where compaction processes and surface melt occur heterogeneously over the variable terrain (Hori et al., 2006; Hulley et al., 2014; Shea and Jamieson, 2011). Therefore, the icefields undergo disparate changes in emissivity over hours to days, meaning that identifying a single representative emissivity value is challenging. Employing too high an emissivity value in the calculation of LST would result in too low a surface temperature. Finally, during the production of MODIS LSTs, clouds and blowing snow can produce low temperatures if they are erroneously categorized as the land surface (Westermann et al., 2012). Without accurate cloud masking, offsets in MODIS LSTs have been previously observed at Summit, Greenland in both summer ($\sim$3°C; Koenig and Hall, 2010) and winter ($\sim$5°C; Shuman et al., 2014). However, the cloud mask has since been updated to address this problem (Yao et al.,

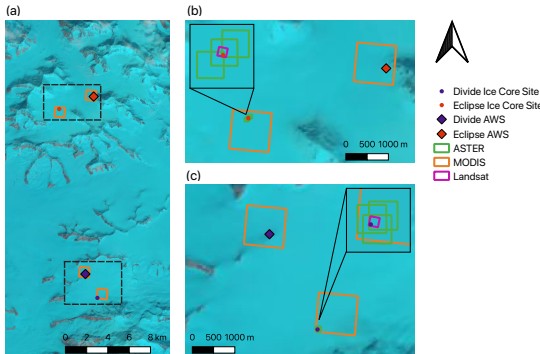

**Figure 2.** MODIS (orange box), ASTER (green box), and Landsat (pink box) footprints at Eclipse and Divide ice core and AWS sites. a) Upper Kaskawulsh-Donjek area containing Eclipse and Divide sites, b) Eclipse ice core (red dot) and AWS (red diamond) sites, c) Divide ice core (blue dot) and AWS (blue diamond) sites. Dashed lines in panel (a) show extents of panels (b) and (c). Both the Divide AWS sensors (Campbell 107F and HOBO-S-THB-M008) were located at the Divide AWS site. Both the Eclipse AWS and iButton sensor were located at the Eclipse AWS site. Background imagery from Landsat 8 on June 30, 2017.

2020). Additionally, cloudy conditions tend to correlate with warmer temepratures, and previous work in the St. Elias indicates that MODIS LSTs over warm (>0°C) surfaces are an average of <2°C higher than corresponding air temperatures (Walsh and Chapman, 1998; Williamson et al., 2013). We therefore do not address the MODIS cloud mask in this study.

Our goal in this study is to determine whether the dominant source of the offset in MODIS LSTs from AWS temperatures at glaciated sites in the heterogeneous alpine environment of the St. Elias arises from (a) the large spatial footprint of the MODIS sensor in highly heterogeneous alpine terrain, (b) poorly constrained snow emissivity values, or (c) a real temperature difference between the surface and air due to near-surface temperature inversions. Since prior work has been unable to fully evaluate the MODIS cold offset in alpine environments due to data limitations, the relative importance of competing hypotheses is unknown.

Additionally, near-surface temperature inversions have to date only been studied on the major ice sheets, and their applicability to alpine environments remains untested. Here, we use two decades of overlapping MODIS and AWS measurements from the St. Elias to resolve some of these uncertainties and develop a surface-to-air conversion factor for use in similar environments that lack AWS data. Although MODIS LSTs can be a useful complement to in situ air temperatures, the two cannot be directly compared and physical differences between the two must be accounted for when using them together. The AWS record from the

St. Elias is, to our knowledge, the longest such record from a glaciated high alpine area in Alaska and the surrounding region. This work is therefore novel in its pairing of a uniquely long AWS temperature record with MODIS LSTs in an understudied system (glaciated high alpine regions) where we often rely solely on remote sensing data for temperature information, as well as in a location with severe consequences in terms of ice mass loss.

**Table 1.** Temperatures used in this study. Type refers to air or surface temperature. Technique refers to the instrumentation or method of measurement. Unknown values are left blank.

| Name | Type | Technique | Footprint | Product/instrument | Uncertainty |
|---|---|---|---|---|---|
| Divide AWS | Air | In situ | Point measurement | Campbell 107F | ± 0.2 °C |
| | | | | HOBO S-THB-M008 12-bit sensor | ± 0.21 °C |
| Eclipse AWS | Air | In situ | Point measurement | | |
| iButton | Air | In situ | Point measurement | Maxim Integrated iButton Data Logger DS1922L | ± 0.5 °C |
| MODIS LST | Surface | Remote sensing | 1 km | MYD21 | |
| ASTER surface temperature | Surface | Remote sensing | 90 m | AST_08 | |
| MODIS BT | Surface | Remote sensing | 1 km | MODTBGA_006 | |
| ASTER BT | Surface | Remote sensing | 90 m | Calculated from ASTL1T following Ndossi and Avdan (2016) | |
| Landsat BT | Surface | Remote sensing | Resampled from 100 m to 30 m (Landsat 8) | LC08 | |
| | | | Resampled from 120 m to 30 m (Landsat 5, 7) | LE07, LT05 | |

## 2 Methods

### 2.1 Study sites and in situ air temperature data

In situ and MODIS temperature data were collected at study sites Eclipse Icefield (60.84° N, 139.84° W, 3,017 m a.s.l.; hereafter referred to as "Eclipse"), and Icefield Divide (60.68° N, 139.78° W, 2,603 m a.s.l.; hereafter referred to as "Divide"; Fig. 1) in the St. Elias Mountains. Instrumentation for both remote sensing and in situ temperatures used here is discussed below. A summary of the temperatures used in this study is shown in Table 1. Surface melt is present but limited at both sites, which are situated in the accumulation zone. Surface melt at these sites does not result in standing surface water, but rather saturates or percolates below the surface, limiting its effect on surface albedo. Observed early melt season (May/June) surface conditions were a fairly soft and flat snow surface with no sastrugi, drifting, or other surface features.

In situ temperatures at Divide were obtained from two adjacent AWS located on small nunataks, the first of which used a Campbell 107F temperature probe (± 0.2°C) housed inside a solar radiation shield, which recorded hourly readings from 2002-2015. The second AWS was located ~300 m from the first, and recorded hourly temperatures with a HOBO S-THB-M008

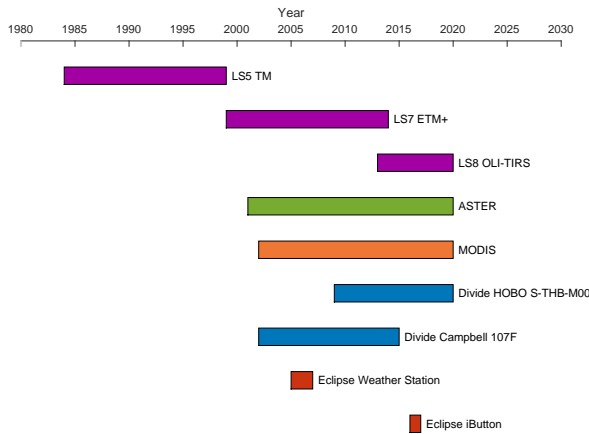

**Figure 3.** Temporal coverage of datasets used in this study. Time periods covered are 1984-present (LS5 TM, LS7 ETM+ and LS8 OLI-TIRS; together referred to as "Landsat"), 2001-present (ASTER), 2002-present (Divide HOBO S-THB-M008 and Divide Campbell 107F; together referred to as "Divide AWS"), 2002-present (MODIS), 2005-2007 (Eclipse weather station), and 2016-2017 (Eclipse iButton). The Eclipse weather station and iButton are together referred to "Eclipse AWS".

12-bit sensor ($\pm$ 0.21°C) housed inside a solar radiation shield from 2009-present (Fig. 3). Both sensors collected temperature data as hourly averages of 5 minute sampling intervals (Williamson et al., 2020). In the window where the two sensors at Divide overlap, we use the HOBO S-THB-M008 12-bit sensor because it provides contemporary solar radiation, relative humidity, wind speed, and pressure data. Both sensors at Divide were located ∼2 m above the surface. The height of sensors above the

surface changed with snow accumulation; however, accumulation on nunataks at Divide is typically limited by intense wind scouring so the sensor height above the surface remains relatively constant over time. Available temperature data at Eclipse are lower quality than at Divide, with limited temporal coverage and sensors not up to World Meteorological Organization standards. We therefore focus on data from Divide, but include available data from Eclipse with the caveat that results are less robust. Around 88% of the temperature data used in this study came from Divide. Additionally our examination of other

meteorological variables and our surface energy balance calculations are all performed with data from Divide.

    Temperatures at Eclipse were obtained from an AWS from 2005-2007, and a Maxim Integrated iButton Data Logger DS1922L ($\pm$ 0.5°C) from 21 May 2016 to 17 May 2017, both located on or near a bedrock outcrop ∼3 km from the site of an ice core drilled at Eclipse in 2016 (Fig. 3). The AWS recorded hourly averages of 5 minute sampling intervals using digital sensors housed in a passively vented radiation shield at a height of approximately 2 m (Williamson et al., 2020). The

iButton recorded temperatures at 3-hour intervals and was placed inside an unvented clear plastic container shielded with rocks. Because data is so limited at Eclipse, we combine the AWS and iButton datasets for maximum coverage at the site. We refer to both the Divide AWS and the combined Eclipse iButton and AWS data as "AWS" for the remainder of this paper.

## 2.2 MODIS LST data

In this assessment of possible sources for the MODIS LST offset, we use the MODIS MYD21 v006 LST product. The MOD21
and MYD21 (together referred to as MxD21) products dynamically retrieve emissivity values for each grid cell, rather than
assigning them based on land cover as was done for the MxD11 products previously examined (Williamson et al., 2017;
McConnell, 2019), and have been shown to correct for MxD11 cold offsets over barren, but not glaciated, surfaces (Hulley,
2017; Li et al., 2020; Yao et al., 2020). MOD21 LSTs were not included in this study as the product was discontinued due to
an optical crosstalk issue in the infrared bands (Hulley, 2017), therefore we focus solely on MYD21 LST data.

Our goal is to determine the dominant source of the offset in MODIS LSTs at glaciated sites in the St. Elias. Because the
Eclipse and Divide AWS are located on nunataks, we test for the LST offset using MODIS data encompassing adjacent ice
core sites ∼3 km from each AWS location, thereby excluding the dark nunatak surface from the MODIS grid cell and focusing
on the ice surface (Fig. 2). We compute the difference in MODIS LST between the ice core site grid cell (containing only ice)
and the AWS site grid cell (containing ice and rock) to determine whether the inclusion of the nunatak has a discernible effect
on the MODIS LST.

MODIS LST data were obtained for the period 2000-2020 (https://lpdaacsvc.cr.usgs.gov/appeears/) for dates with minimal
cloud cover between the hours of 12:00 and 13:00 (local solar time), when viewing angle is < 30°, to mitigate the effect of
viewing angle on temperature and emissivity. LSTs with an error of >1°C were excluded from this study. At Divide, 742
MODIS images spanning 2002-2020 were analyzed. Seasonally, 203 images were acquired in spring (MAM), 169 in summer
(JJA), 188 in fall (SON), and 182 in winter (DJF). At Eclipse,100 MODIS images were analyzed: 87 spanning June 2005
through June 2007 and 13 spanning November 2016 through February 2017. Each MODIS image was paired with the closest
hourly measurement available in the AWS data. MODIS LSTs were subtracted from the nearest hourly in situ air temperature
measurement to calculate their offset from in situ temperatures. A small number of summer MODIS LST offset results were
skewed by air temperatures well above 0°C (30 dates with air temperature > 4°C, 5 dates with air temperature >8°C), as the
snow surface cannot warm above freezing without melting. Removing these data reduced the temporal coverage of the summer
MODIS LST offset data, but had no effect on the seasonal distribution of the cold offset. All LSTs above freezing were retained,
as none exceeded the 1°C measurement error.

## 2.3 Sensor footprint size

To test if the LST offset is a result of the MODIS sensor's large footprint, we calculate the offset of both ASTER (90 m footprint)
and MODIS (1 km footprint) surface temperatures from AWS measurements and then compare the magnitude of the offsets.
We use only ASTER and MODIS images from the 12:00-13:00 window that had paired AWS data in this comparison. ASTER
kinetic temperature data (AST08, https://search.earthdata.nasa.gov/search) for 2001-2020 were manually filtered to remove
dates with cloud cover or inconsistency in their time of acquisition, resulting in 33 ASTER images coincident with MODIS
imagery at Divide, and 15 at Eclipse. The seasonal distribution of acquired ASTER imagery is heavily skewed, with only
three images available during winter months and none during spring. While Landsat also has a smaller footprint than MODIS

(100-120 m), Landsat surface temperatures remain under development (as of July 2021) and were therefore not included in this study.

## 2.4 Snow surface emissivity

To test if the MODIS LST offset is a result of poorly constrained snow emissivity values, we assess whether the prominent wintertime offset in MODIS LSTs is also present in MODIS brightness temperatures prior to the incorporation of snow surface emissivity. MODIS brightness temperatures (https://lpdaacsvc.cr.usgs.gov/appeears/) were extracted, and their offset from AWS temperatures was calculated. We also examine ASTER and Landsat brightness temperatures because of their higher spatial resolution (90 m for ASTER, 100-120 m for Landsat). ASTER brightness temperatures were obtained from TIR imagery (https://search.earthdata.nasa.gov/search; using the methods of Ndossi and Avdan 2016). Landsat top of atmosphere brightness temperature imagery (https://earthexplorer.usgs.gov/) was visually examined for cloud cover, and cloud-free grid cells were extracted for analysis using QGIS.

Additionally, we compare the MODIS LST offset with snow accumulation data from Divide. The Divide accumulation record was obtained using a Campbell Scientific SR50 ultrasonic snow depth sounder instrument. The instrument provided twice-daily readings of its distance from the snow surface at the Icefield Discovery Camp during the period spanning 2003-2012, corrected for the variability in speed of sound with air temperature.

## 2.5 Near-surface temperature inversions

To test whether the MODIS LST offset reflects pervasive near-surface temperature inversions, we examine whether the offset is more pronounced under conditions that facilitate near-surface inversions, namely low levels of incoming solar radiation and low wind speeds. Low solar radiation gives rise to near-surface inversions, but it can be counterbalanced if wind speeds are high enough to disturb thermal stratification (Adolph et al., 2018). During one study in Greenland (at Summit), no inversions greater than 2°C were observed in the 2 m above the snow surface when incoming solar radiation was above $600\,\mathrm{Wm^{-2}}$ or wind speed was greater than approximately $7\,\mathrm{ms^{-1}}$ (Adolph et al., 2018). In another study across 22 sites in Greenland, maximum temperature inversions were observed at wind speeds of 3-5 $\mathrm{ms^{-1}}$ (Nielsen-Englyst et al., 2019). We compare differences between AWS and MODIS LST data to wind speed and solar radiation data obtained from the Divide AWS. We transform LST offsets, wind speed, and solar radiation data to approximately normal distributions using a box-cox transformation and normalize each dataset around zero. We then perform linear regressions on LST offsets vs. wind speed, LST offsets vs. solar radiation, and LST offsets vs. wind speed under low ($<400\,\mathrm{Wm^{-2}}$) levels of solar radiation.

We compare the magnitude of the LST offset to wind speed and solar radiation data obtained from the Divide AWS. To test if a near-surface temperature inversion could occur under surface conditions at Divide and Eclipse, we compare differences in AWS and MODIS temperatures to surface temperatures calculated with the following simple energy balance model. The net surface energy balance ($E_N$) can be expressed by:

$$E_N = E_S \downarrow + E_S \uparrow + E_L \downarrow + E_L \uparrow + E_G + E_H + E_E + E_P \tag{1}$$

where $E_S \downarrow$ is the downward shortwave radiation, $E_S \uparrow$ is the reflected shortwave radiation, $E_L \downarrow$ is the downward longwave radiation, $E_L \uparrow$ is the upward emitted longwave radiation, $E_G$ is the subsurface energy flux, $E_H$ and $E_E$ are the turbulent sensible and latent heat fluxes, and $E_P$ is the heat flux associated with liquid precipitation that subsequently freezes (Cuffey and Paterson 2010). We focus on the radiative fluxes ($E_S \downarrow$, $E_S \uparrow$, $E_L \downarrow$, and $E_L \uparrow$), as our goal is simply to determine whether observed temperature differences are physically plausible, and not to produce a precise energy balance model. We ignore $E_G$ because it is often small relative to both radiative and turbulent fluxes, and several studies (e.g. Brock and Arnold, 2000; Hock and Noetzli, 1997, Favier et al., 2004) have validated energy models in which it is omitted (Hock and Holmgren, 1996; Pellicciotti et al., 2009; Yang et al., 2021). Subsurface energy fluxes have been found to represent only 1-2% of the total heat flux on glacier surfaces (Giesen et al., 2008; Yang et al., 2011). We ignore $E_P$, as rainfall has not been observed in the St. Elias icefields. We also ignore turbulent fluxes, as they are both difficult to calculate and unnecessary for our purposes of evaluating the physical plausibility of observed temperature differences. LST offsets observed in this study are most prominent under low wind speed conditions, when turbulent fluxes are unlikely to be a dominant component of the surface energy balance. Ignoring turbulent fluxes, we can still calculate an upper bound for temperature inversion strength under site conditions at Divide and Eclipse. After applying our simplifying assumptions, equation 1 becomes:

$$E_N \approx E_S \downarrow + E_S \uparrow + E_L \downarrow + E_L \uparrow \tag{2}$$

We assume a net surface energy balance of $E_N = 0$. $E_L \uparrow$ is the energy emitted by the earth's surface and can be described by:

$$E_L \uparrow = \epsilon_s \sigma T_s{}^4 \tag{3}$$

where $\epsilon_s$ is surface emissivity, $\sigma$ is the Stefan-Boltzmann constant, and $T_s$ is surface temperature (Cuffey and Paterson, 2010). Expressing $E_L \uparrow$ in terms of its components, and rearranging to solve for surface temperature, we obtain:

$$T_s \approx \left( \frac{E_L \downarrow + E_S \downarrow (1 - \alpha)}{\epsilon_s \sigma} \right)^{0.25} \tag{4}$$

where $\alpha$ is surface albedo. We acquire downward shortwave radiation from the Divide AWS. We calculate downward longwave radiation as follows, using 2 m air temperature ($T_a$) from Divide and atmospheric emissivity ($\epsilon_a$) from the ERA5 reanalysis product:

$$E_L \downarrow = \sigma \epsilon_a T_a{}^4 \tag{5}$$

We use only the derived emissivity from the ERA5 product, rather than the total downward radiation in order to use measured values (in situ 2 m air temperature) where possible. ERA5 outputs have a spatial resolution of 31 km; data are available every six hours from 2002-2019 (Hersbach et al., 2020). Atmospheric emissivity increases with increasing surface vapor pressure (Staley and Jurica, 1972). Our atmospheric emissivity values ranged from ~0.48 to 1. Atmospheric emissivity measured over the Sierra Nevada (Spain) from 2005-2011 ranged from ~0.4-1 (Herrero and Polo, 2012). Prior work in the St. Elias has demonstrated issues with MODIS albedo values arising from confusion between snow and cloud cover (Williamson et al.,

2016). We therefore avoided using the MODIS albedo product to eliminate this unnecessary source of uncertainty. Instead,
we use a surface albedo of 0.742, which was the mean albedo measured at Divide during August, 2015 (Williamson et al.,
2016). We use end-member snow emissivity values of $\epsilon_s = 0.95$ and $\epsilon_s = 0.99$ (Hori et al., 2006). The MODIS emissivity
values for the days sampled in this study range from 0.930 to 0.988. The range of emissivity values is similar in all seasons,
so we consider distinguishing by season unnecessary for our simple model. The distribution of emissivity values is skewed
toward higher values, so we consider the 0.95 value from Hori et al. (2006) a reasonable choice for our lower emissivity bound.
We assign a value of 0°C to all surface temperatures calculated to be above 0°C because a snow surface cannot exceed this
temperature without melting.

## 2.6 Approximating air temperatures with MODIS LSTs

If air temperature values can be approximated from MODIS LSTs, MODIS LSTs can be used in conjunction with sparse air
temperature measurements to examine climatic conditions associated with physical phenomena of interest such as surface melt
or changes in ice chemistry. Here, we compare interannual trends between the MODIS LSTs and in situ air temperatures, and
reconcile the difference between MODIS LSTs and AWS temperatures using a simple linear regression. We are interested
in interannual trends since interpretation of paleo records often occurs on interannual timescales. We therefore calculate the
mean annual value for both MODIS LSTs and AWS temperatures. Because we calculate annual means rather than examine
individual MODIS LST and AWS temperature pairs, we use all available MODIS LSTs and AWS temperatures, rather than only
265 the subset of dates for which we have both. We fit a linear model to mean annual MODIS LSTs and AWS temperatures using
the MATLAB function fitlm(), taking the AWS temperature to be the response variable. We then use the coefficients from this
linear fit model to approximate a set of mean annual air temperatures from MODIS LSTs ($T_{air,approx.} = -3.73 + 0.44 LST$).
The RMSE of our linear fit is 1.9°C, and the interannual variability spans a range of 5.3°C.

## 3 Results

### 3.1 MODIS LSTs at Divide ice core and AWS sites

MODIS data at the Divide AWS nunatak and adjacent ice core site have a median temperature difference of 0.8°C and in-
terquartile range of 2.0 °C (Table 2). The difference between the two sites shows greater variability in the fall (IQR = 3.2°C)
and winter (IQR = 4.0°C) than in the spring (IQR = 0.7°C) and summer (IQR = 1.8°C), with the ice core site tending to be
slightly colder in the winter (median temperature difference of -0.5°C), but warmer in the spring (Mdn = 1.0°C), summer (Mdn
= 1.3°C) and fall (Mdn = 0.3°C; Fig. 4).

### 3.2 Seasonal distribution of the MODIS LST offset

In comparing MODIS LSTs with AWS temperatures at Divide and Eclipse, we find the MODIS LSTs to be slightly lower that
coincident AWS temperatures, with the offset to be greatest during the fall and winter (Fig. 5; Table 3). We report a warmer

**Table 2.** Median differences between MODIS LSTs at the Divide ice core and AWS sites, median MODIS LST offsets by season at Divide and Eclipse, median MODIS brightness temperature (BT) offsets by season at Divide and Eclipse, and median calculated temperature inversions with surface emissivities of 0.95 and 0.99. Differences between MODIS LSTs at the ice core site and AWS site are reported as ice core site − AWS site (°C). All offsets are reported as MODIS − AWS (°C). Brightness temperatures for bands 31 and 32 are averaged together. Inversions are reported as negative values.

| | Ice Core Site − AWS Site (°C) | MODIS LST − AWS (°C) | | MODIS BT − AWS (°C) | | $T_{surface} - T_{air}$ (°C) | |
| --- | --- | --- | --- | --- | --- | --- | --- |
| Season | Divide | Divide | Eclipse | Divide | Eclipse | $\epsilon_s = 0.95$ | $\epsilon_s = 0.99$ |
| Spring (MAM) | 0.3 | −0.7 | −1.7 | −1.7 | −2.8 | 8.0 | 7.5 |
| Summer (JJA) | 1.3 | −1.0 | −1.1 | −2.4 | −2.5 | 0.7 | 0.7 |
| Fall (SON) | 0.3 | −4.4 | −5.2 | −5.6 | −5.7 | 2.3 | 1.0 |
| Winter (DJF) | −0.5 | −8.4 | −8.9 | −9.4 | −9.4 | −7.1 | −9.7 |

**Table 3.** Results for Wilcoxon rank sum tests between seasonal MODIS LST offsets at Divide (a) and Eclipse (b). Bolded cells indicate a more pronounced cold offset in the column season, italicized cells indicate a more pronounced cold offset in the row season, and standard font cells indicate no significant difference between the seasons.

(a)

| DIVIDE | Spring (MAM) | Summer (JJA) | Fall (SON) | Winter (DJF) |
| --- | --- | --- | --- | --- |
| *Spring (MAM)* | | z = 0.81<br>p > 0.5 | **z = 9.85**<br>**p < 0.5** | **z = 13.41**<br>**p < 0.5** |
| *Summer (JJA)* | z = 0.81<br>p > 0.05 | | **z = 8.80**<br>**p < 0.05** | **z = 12.35**<br>**p < 0.05** |
| *Fall (SON)* | z = 9.85<br>p < 0.05 | z = 8.80<br>p < 0.05 | | **z = 5.53**<br>**p < 0.05** |
| *Winter (DJF)* | z = 13.41<br>p < 0.05 | z = 12.35<br>p < 0.05 | z = 5.53<br>p < 0.05 | |

(b)

| ECLIPSE | Spring (MAM) | Summer (JJA) | Fall (SON) | Winter (DJF) |
| --- | --- | --- | --- | --- |
| *Spring (MAM)* | | z = 1.11<br>p > 0.5 | **z = 2.62**<br>**p < 0.5** | **z = 3.59**<br>**p < 0.5** |
| *Summer (JJA)* | z = 1.11<br>p > 0.05 | | **z = 3.73**<br>**p < 0.05** | **z = 4.39**<br>**p < 0.05** |
| *Fall (SON)* | z = 2.62<br>p < 0.05 | z = 3.73<br>p < 0.05 | | z = 1.15<br>p > 0.05 |
| *Winter (DJF)* | z = 3.59<br>p < 0.05 | z = 4.39<br>p < 0.05 | z = 1.15<br>p > 0.05 | |

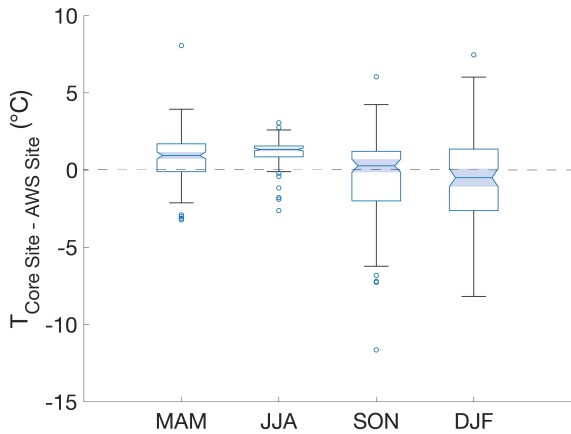

**Figure 4.** Differences between MODIS LSTs at Divide ice core and AWS sites. Shaded notched areas indicate the 95% confidence interval for the median temperature difference. Horizontal dashed line indicates where MODIS LSTs at the ice core site and AWS site are equivalent.

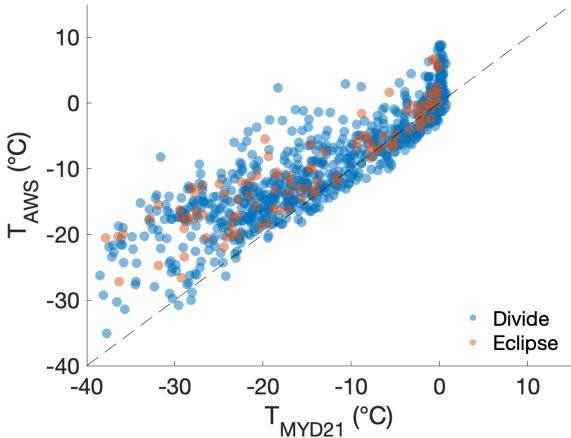

**Figure 5.** MODIS LST vs. air temperatures (AWS) at Divide (blue) and Eclipse (orange). The dashed line indicates where MODIS LST and AWS temperatures are equivalent.

surface as a positive difference and a colder surface as a negative difference. The difference between AWS temperatures and
MODIS LSTs at Divide are larger in the fall (Mdn = $-4.4°$C) and winter (Mdn = $-8.4°$C) than in the spring (Mdn = $-0.7°$C)
and summer (Mdn = $-1.0°$C; Table 2). Winter LST offsets are significantly larger than those in spring, summer, and fall. Fall
LST offsets are significantly larger than those in spring and summer. Differences between AWS temperatures and MODIS
LSTs at Eclipse are also larger in the fall (Mdn = $-5.2°$C) and winter (Mdn = $-8.9°$C) than in the spring (Mdn = $-1.7°$C)
and summer (Mdn = $-1.1°$C). Fall and winter LST offsets do not differ significantly from each other in magnitude. Fall LST

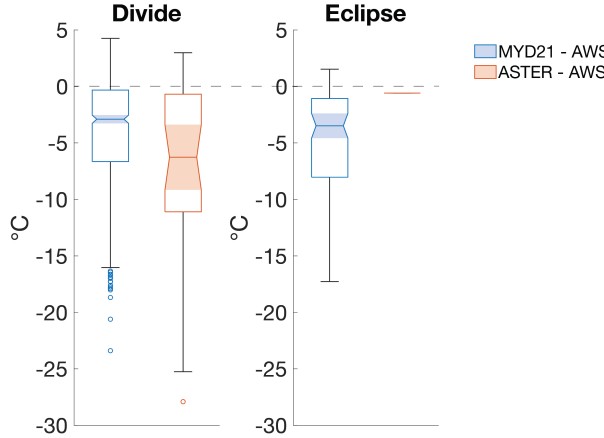

**Figure 6.** Differences between remote sensing surface temperatures and AWS measurements at Divide and Eclipse. Horizontal dashed line indicates where remote sensing temperatures and AWS temperatures are equivalent. Temperature products from both MODIS (a) and ASTER (b) show a cold offset relative to AWS temperatures. Shaded notched areas indicate the 95% confidence interval for the median temperature difference.

offsets are significantly larger than those during spring and summer. Winter LST offsets are likewise significantly larger than those during spring and summer.

### 3.3 Sensor footprint size

In comparing MODIS (1 km) and ASTER (90 m) surface temperatures, we find that they both show an offset relative to AWS measurements at Divide, with the MODIS offset (Mdn = $-2.9$°C) being significantly smaller than the ASTER offset (Mdn = $-6.3$°C; Fig. 6). In all seasons, observed MODIS offsets vary by more than 10°C, with the range of winter values being greatest at 35.6°C at Divide and 25.1°C at Eclipse. No ASTER temperatures were produced coincident with MODIS LSTs during the winter, and only three during the spring so we were unable to bin ASTER data by season. Only one ASTER temperature was produced coincident with MODIS LSTs at Eclipse.

### 3.4 Snow surface emissivity

In comparing MODIS temperature products before and after the incorporation of snow surface emissivity, MODIS brightness temperatures in bands 31 and 32 (prior to the incorporation of snow emissivity) show similar offset patterns as the LST products (after the incorporation of snow emissivity), with the cold offset being most prominent in fall and winter (Table 2, Table 4, Fig. 7). At Divide, winter offsets across both bands (Mdn = $-9.4$°C) are significantly larger than those in spring (Mdn = $-1.7$°C), summer (Mdn = $-2.4$°C), and fall (Mdn = $-5.6$°C). Fall offsets are significantly larger than those in spring and summer. At Eclipse, fall (Mdn = $-5.7$°C) and winter (Mdn = $-9.4$°C) offsets do not differ significantly from each other in magnitude.

**Table 4.** Results for Wilcoxon rank sum tests between seasonal MODIS brightness temperature offsets from AWS temperatures at Divide (a) and Eclipse (b). Brightness temperatures for bands 31 and 32 are averaged together. Bolded cells indicate a more pronounced cold offset in the column season, italicized cells indicate a more pronounced cold offset in the row season, and standard font cells indicate no significant difference between the seasons.

(a)

| DIVIDE | Spring (MAM) | Summer (JJA) | Fall (SON) | Winter (DJF) |
|---|---|---|---|---|
| *Spring (MAM)* | | **z = 3.60** <br> **p < 0.5** | **z = 13.39** <br> **p < 0.5** | **z = 19.09** <br> **p < 0.5** |
| *Summer (JJA)* | *z = 3.60* <br> *p < 0.05* | | **z = 10.20** <br> **p < 0.05** | **z = 16.70** <br> **p < 0.05** |
| *Fall (SON)* | *z = 13.39* <br> *p < 0.05* | *z = 10.20* <br> *p < 0.05* | | **z = 8.63** <br> **p < 0.05** |
| *Winter (DJF)* | *z = 19.09* <br> *p < 0.05* | *z = 16.70* <br> *p < 0.05* | *z = 8.63* <br> *p < 0.05* | |

(b)

| ECLIPSE | Spring (MAM) | Summer (JJA) | Fall (SON) | Winter (DJF) |
|---|---|---|---|---|
| *Spring (MAM)* | | z = 0.95 <br> p > 0.5 | **z = 3.25** <br> **p < 0.5** | **z = 4.70** <br> **p < 0.5** |
| *Summer (JJA)* | z = 0.95 <br> p > 0.05 | | **z = 4.57** <br> **p < 0.05** | **z = 5.83** <br> **p < 0.05** |
| *Fall (SON)* | *z = 3.25* <br> *p < 0.05* | *z = 4.57* <br> *p < 0.05* | | z = 1.77 <br> p > 0.05 |
| *Winter (DJF)* | *z = 4.70* <br> *p < 0.05* | *z = 5.83* <br> *p < 0.05* | z = 1.77 <br> p > 0.05 | |

Fall offsets are significantly larger than those during spring (Mdn = $-2.8°C$) and summer (Mdn = $-2.5°C$). Winter offsets are likewise significantly larger than those during spring and summer.

Landsat brightness temperatures at Divide also show a pattern of greater offset from AWS temperatures in the fall (Mdn = $-4.2°C$) and winter (Mdn = $-12.1°C$) than in the spring (Mdn = $-1.3°C$) and summer (Mdn = $-2.7°C$). Winter offsets are significantly larger than those in spring, summer, and fall. Fall offsets are significantly larger than those in spring and summer.

Regarding emissivity changes associated with snowfall events, we find no relationship either between the LST offset and individual snowfall events or between the LST offset and the total accumulation each month, the percent of days with accumulation each month, or the mean days between accumulation each month. We also find no relationship between the LST offset and days since last accumulation.

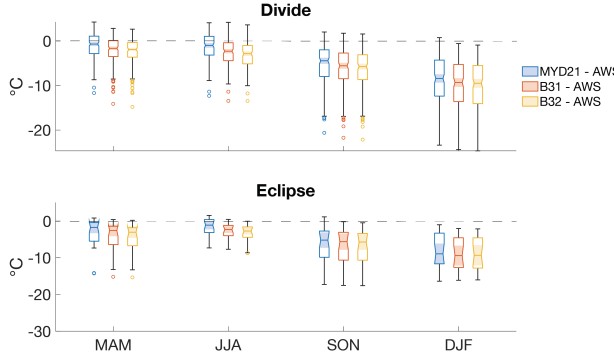

**Figure 7.** Offsets of MODIS surface temperatures, MODIS Band 31 brightness temperatures and MODIS Band 32 brightness temperatures from AWS measurements at Divide and Eclipse. Horizontal dashed lines indicate where MODIS temperatures and AWS temperatures are equivalent. At Divide, spring and summer offsets are smaller in the final surface temperatures than in brightness temperatures (95% confidence interval); fall and winter offsets no difference between final surface temperatures and brightness temperatures (95% confidence interval). Surface and brightness temperatures show no significant difference from each other at Eclipse in any season due to smaller sample sizes.

## 3.5 Near-surface temperature inversions

Similar to findings at Summit, Greenland (Adolph et al., 2018), the MODIS LST offset in the St. Elias is most pronounced under conditions that facilitate near-surface temperature inversions, namely low wind speeds and low levels of incoming solar radiation (Fig. 8). The magnitude of the offset correlates weakly with wind speed ($r^2 = 0.02$, $p < 0.05$) and more strongly with solar radiation ($r^2 = 0.35$, $p < 0.05$; Fig. 9). Nearly all (97%) MODIS LST offsets in excess of 10°C are coincident with solar radiation lower than 430 W m$^{-2}$. An overwhelming majority (95%) of MODIS LST offsets in excess of 10°C are coincident with wind speeds lower than 40 km h$^{-1}$. Comparing these findings with modeled results, we find that modeled temperature inversions are also strongest in the winter. Modeled surface temperatures show a more pronounced offset from 2 m air temperatures in winter than in spring, summer, and fall (Table 5). The observed median MODIS LST offset is 8.4°C in the winter and 1.0°C in the summer (Table 2). Our simple energy balance model predicts a median temperature inversion of 4.8°C ($\epsilon_s = 0.95$) and 7.4°C ($\epsilon_s = 0.99$) in the winter, and no inversion in the summer (Fig. 10). In the winter, modeled inversion strength varies by up to 60°C. In the summer, these data show a much narrower spread because of our 0°C cap on surface temperatures. The diurnal surface temperature offset cycle is more pronounced in the summer than in the winter, with the greatest offset occurring during nighttime hours, justifying the decision to limit MODIS LSTs to midday image collection (Fig. 11; Tables 6, 7).

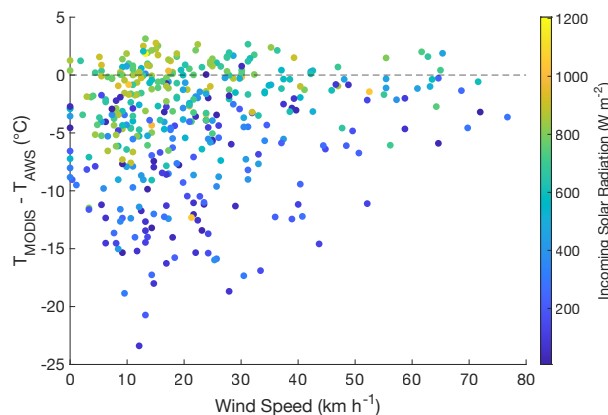

**Figure 8.** Comparison of the MODIS LST offset (MODIS-AWS) with measured solar radiation and wind speed at Divide. The MODIS LSTs show the most pronounced cold offset at low levels of solar radiation (shown by marker color) and low wind speeds. Horizontal dashed line marks all locations where MODIS = AWS.

### 3.6 Approximating air temperatures from MODIS LSTs

Interannual trends in MODIS LSTs agree well with those in AWS temperatures ($r^2 = 0:23$ and $p < 0.05$; Fig. 12). Our simple linear regression ($T_{air,approx.} = -3.73 + 0.44 LST$) reconciles the difference between mean annual MODIS LSTs and AWS temperatures (mean error of $0.0 \pm 1.8°$C) for individual years.

## 4 Discussion

### 4.1 MODIS LSTs at Divide ice core and AWS sites

MODIS LSTs at the ice core site do not tend to be colder than at the AWS site except during the winter. The inclusion of the warmer nunatak surface in the MODIS grid cell at the AWS site fails to provide a compelling explanation for the colder wintertime LSTs at the ice core site, given that more of the rock surface would likely have snow cover during the winter. The colder wintertime LSTs at the ice core site may contribute to the MODIS LST offset from in situ temperature measurements examined in this study. However, this contribution is too small (median = -0.5°C) to explain the magnitude of the MODIS LST offset at the Divide ice core site (median = -8.4°C). In the spring, summer, and fall, the LSTs at the ice core site tend to be slightly warmer than at the AWS site. Results here may therefore underestimate the magnitude of the MODIS LST offset from AWS temperatures in these seasons.

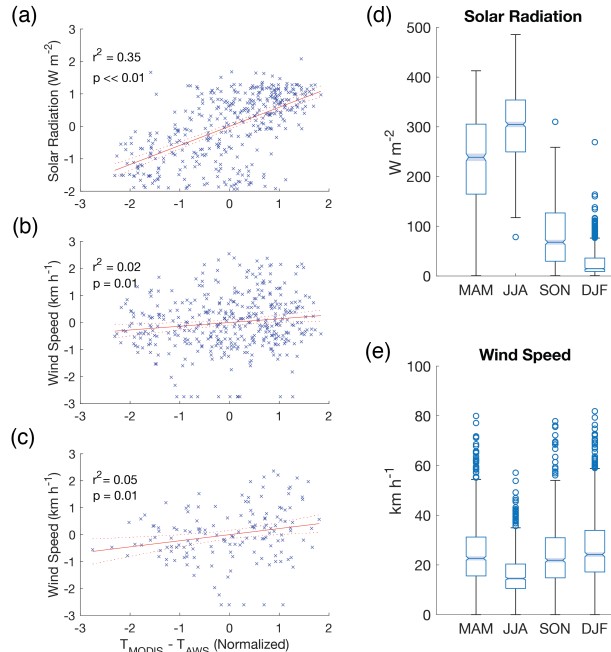

**Figure 9.** Linear regressions of the normalized MODIS LST offset vs. solar radiation (a) wind speed (b), wind speed under low (< 400 $Wm^{-2}$) solar radiation conditions (c), and boxplots of solar radiation (d) and wind speed (e) by season. The magnitude of the MODIS LST offset is more strongly related to solar radiation than to wind speed. Dashed red lines in regression plots indicate the 95% confidence interval around the regression line. Notches and shading in boxplots indicate the upper and lower bounds of each season's median value of solar radiation or wind speed at the 95% confidence interval.

## 4.2 Sensor footprint size

Despite ASTER's smaller footprint and the homogeneity of surface type within its grid cell relative to that within the MODIS grid cell, the LST offset persists in ASTER data (Fig. 6). The LST offset in ASTER data indicates that MODIS LSTs do not display an offset from AWS temperatures simply because they are mean temperatures over square kilometer grid cells rather than point measurements. Additionally, AWS temperatures at Divide and Eclipse show good coherence, with a mean temperature difference between the sites of $0.9 \pm 2.0°C$, despite the two sites being 30 km apart and over 400 m of elevation

difference between them. At its most extreme, the temperature difference measured by weather stations between the two sites reaches ∼8°C. Although 8°C is notable, the fact that it is on the upper extreme of temperature disparities over 30 km and 400 m of elevation demonstrates that averaging temperatures over a single square kilometer is unlikely to routinely produce

**Table 5.** Results for Wilcoxon rank sum tests between modeled temperature inversions by season. Inversions ($T_{surface} - T_{air}$) were calculated from ERA5 and Divide AWS data. Bolded cells indicate a larger inversion in the column season, italicized cells indicate a larger inversion in the row season, and standard font cells indicate no significant difference between the seasons.

(a)

| $\epsilon = 0.95$ | Spring (MAM) | Summer (JJA) | Fall (SON) | Winter (DJF) |
|---|---|---|---|---|
| *Spring (MAM)* | | **z = 27.84** **p < 0.5** | **z = 17.41** **p < 0.5** | **z = 21.81** **p < 0.5** |
| *Summer (JJA)* | *z = 27.84* *p < 0.05* | | *z = 9.95* *p < 0.05* | **z = 10.75** **p < 0.05** |
| *Fall (SON)* | *z = 17.41* *p < 0.05* | **z = 9.95** **p < 0.05** | | **z = 11.82** **p < 0.05** |
| *Winter (DJF)* | *z = 21.81* *p < 0.05* | *z = 10.75* *p < 0.05* | *z = 11.82* *p < 0.05* | |

(b)

| $\epsilon = 0.99$ | Spring (MAM) | Summer (JJA) | Fall (SON) | Winter (DJF) |
|---|---|---|---|---|
| *Spring (MAM)* | | **z = 26.81** **p < 0.5** | **z = 18.76** **p < 0.5** | **z = 23.96** **p < 0.5** |
| *Summer (JJA)* | *z = 26.81* *p < 0.05* | | *z = 5.62* *p < 0.05* | **z = 15.38** **p < 0.05** |
| *Fall (SON)* | *z = 18.76* *p < 0.05* | **z = 5.62** **p < 0.05** | | **z = 13.29** **p < 0.05** |
| *Winter (DJF)* | *z = 23.96* *p < 0.05* | *z = 15.38* *p < 0.05* | *z = 13.29* *p < 0.05* | |

an offset of similar magnitude in wintertime MODIS LSTs (Mdn = $-8.4$°C at Divide, Mdn = $-8.9$°C at Eclipse). MODIS' footprint size is thus not the dominant source of the offset in its LSTs.

## 4.3 Snow surface emissivity

We find similar seasonal distributions of offset from AWS temperatures in MODIS LSTs and MODIS brightness temperatures, suggesting that the preferential fall and winter offset is not introduced by the conversion from brightness temperature to surface temperature or the emissivity values used in this conversion (Fig. 7). Moreover, Landsat brightness temperatures also show a pattern of greater offset from AWS temperatures in the fall and winter. The observed cold offset in MODIS LSTs is therefore not unique to the MYD21 product or even the MODIS sensor. Unfortunately, due to the limited availability of ASTER data, too few images exist to examine any seasonal pattern.

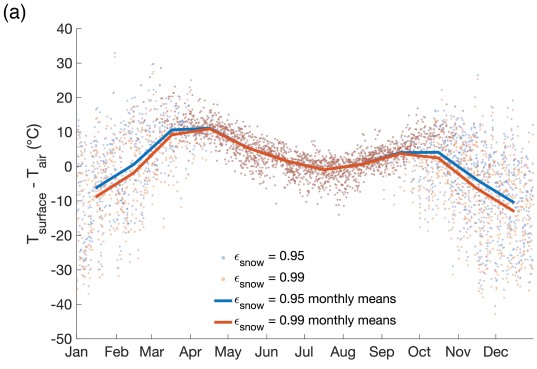

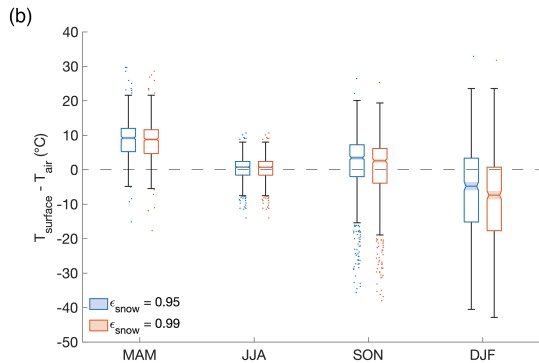

**Figure 10.** Seasonal differences between surface and 2 m air temperatures calculated from ERA5 and Divide AWS data. All data are for time 12:00 to control for diurnal effects. Shaded notched areas in panel (b) indicate the 95% confidence interval for the median temperature difference. Horizontal dashed line in panel (b) indicates where surface temperatures and air temperatures are equivalent. All surface temperatures $> 0°\text{C}$ were assigned a value of $0°\text{C}$.

While results here show that poorly constrained emissivity values do not introduce the cold offset, they may exacerbate it. Applying an accurate emissivity correction to MODIS brightness temperatures should bring the resultant surface temperatures closer to AWS measurements. At Divide, MODIS surface temperatures are $\sim 60\%$ closer to AWS measurements than MODIS brightness temperatures during spring and summer (significant at the 95% confidence interval, Fig. 7). During the fall and winter, however, there is no significant difference between the median offsets in MODIS brightness and surface temperatures (95% confidence interval), suggesting that emissivity values during these seasons may contribute to the offset in resultant surface temperatures. At Eclipse, the median offset between MODIS LSTs and AWS temperatures does not differ from that between MODIS brightness and AWS temperatures in any season (95% confidence interval). However, Eclipse imagery was limited (20-30 samples per season at Eclipse vs. 169-203 samples per season at Divide), so a robust analysis could not be completed.

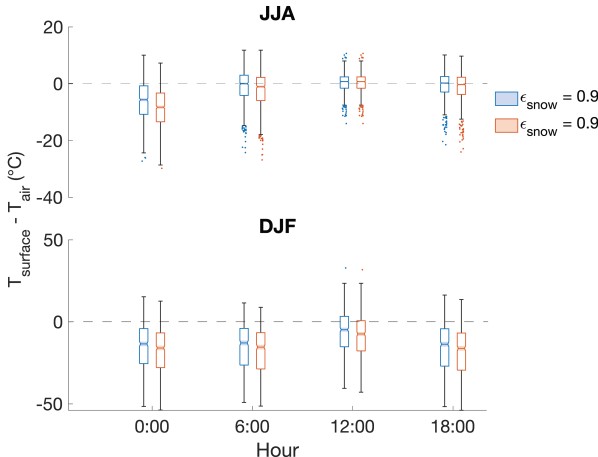

**Figure 11.** Diurnal differences between surface and 2 m air temperatures calculated from ERA5 and Divide AWS data. Summer (JJA) and winter (DJF) data are shown separately to control for seasonal effects. Shaded notched areas indicate the 95% confidence interval for the median temperature difference. Horizontal dashed lines indicate where surface temperatures and air temperatures are equivalent. All surface temperatures > 0°C were assigned a value of 0°C.

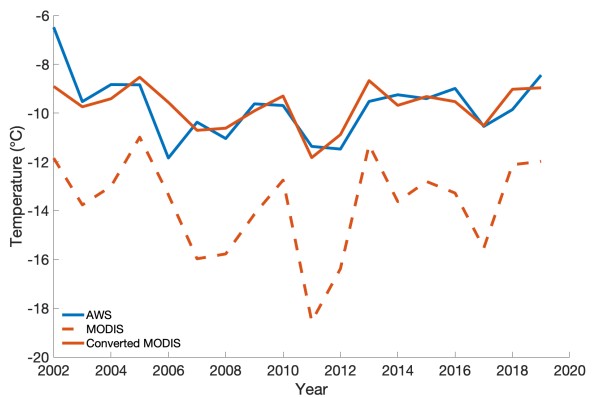

**Figure 12.** AWS temperatures, MODIS LSTs, and MODIS LSTs converted to air temperatures at Divide. Unconverted MODIS LSTs (dashed orange line) show a prominent offset from AWS measurements, but overall agreement in years of high vs. low temperatures. MODIS LSTs converted to air temperatures by applying a simple linear regression (solid orange line) show much closer agreement with AWS temperatures (solid blue line). All temperatures are mean annual values.

Emissivity values may be especially poorly known under winter conditions because of rapidly changing snow surface characteristics during and after snowfall events, resulting in the seasonal difference in outcome of the LST algorithm as seen at Divide. Emissivity increases with surface melt, and decreases with increasing particle size and density, which can occur due

**Table 6.** Results for Wilcoxon rank sum tests between modeled temperature inversions at hours 00:00, 06:00, 12:00, and 18:00 during the summer using an emissivity value of $\epsilon = 0.95$ (a) and $\epsilon = 0.99$ (b). Inversions ($T_{surface} - T_{air}$) were calculated from ERA5 and Divide AWS data. Bolded cells indicate a larger inversion in the column hour, italicized cells indicate a larger inversion in the row hour, and standard font cells indicate no significant difference between the hours.

(a)

| $\epsilon = 0.95$ | 00:00 | 06:00 | 12:00 | 18:00 |
|---|---|---|---|---|
| *00:00* | | *z = 16.77* *p < 0.5* | *z = 22.04* *p < 0.5* | *z = 18.73* *p < 0.5* |
| *06:00* | **z = 16.77** **p < 0.05** | | *z = 3.89* *p < 0.05* | z = 0.71 p > 0.05 |
| *12:00* | **z = 22.04** **p < 0.05** | **z = 3.89** **p < 0.05** | | **z = 3.46** **p < 0.05** |
| *18:00* | **z = 18.74** **p < 0.05** | z = 0.71 p > 0.05 | *z = 3.46* *p < 0.05* | |

(b)

| $\epsilon = 0.99$ | 00:00 | 06:00 | 12:00 | 18:00 |
|---|---|---|---|---|
| *00:00* | | *z = 19.30* *p < 0.5* | *z = 28.52* *p < 0.5* | *z = 23.48* *p < 0.5* |
| *06:00* | **z = 19.30** **p < 0.05** | | *z = 8.47* *p < 0.05* | *z = 3.00* *p < 0.05* |
| *12:00* | **z = 28.52** **p < 0.05** | **z = 8.47** **p < 0.05** | | **z = 6.09** **p < 0.05** |
| *18:00* | **z = 23.48** **p < 0.05** | **z = 3.00** **p < 0.05** | *z = 6.09* *p < 0.05* | |

to either packing or welding of grains as the snow surface evolves following a snowfall event (Salisbury et al., 1994). In the 10.5-12.5 μm wavelength range (MODIS bands 31 and 32), emissivity can vary from 0.949 to 0.997 depending on the surface type (fine dendrite snow, medium granular snow, coarse grain snow, sun crust, and bare ice), with lower emissivity values for coarse grain snow and ice than for fine dendrite snow (Wan and Zhang, 1999; Hori et al., 2006). At Divide, summertime emissivity changes are likely dominated by alteration of the surface snow by melt, while wintertime emissivity changes are likely dominated by snow surface evolution following snowfall, which occurs more frequently in the winter. The relative magnitude of summer and winter emissivity changes are unknown and may result in the seasonal difference in outcome of the LST algorithm. However, given the low temporal resolution of the MODIS data relative to the Divide accumulation record (1 image per day vs. 1 sample per hour), we find no relationship between the LST offset and snow accumulation at Divide. Additional sampling is needed to fully evaluate this relationship.

**Table 7.** Results for Wilcoxon rank sum tests between modeled temperature inversions at hours 00:00, 06:00, 12:00, and 18:00 during the winter using an emissivity value of $\epsilon = 0.95$ (a) and $\epsilon = 0.99$ (b). Inversions ($T_{surface} - T_{air}$) were calculated from ERA5 and Divide AWS data. Bolded cells indicate a larger inversion in the column hour, italicized cells indicate a larger inversion in the row hour, and standard font cells indicate no significant difference between the hours.

(a)

| $\epsilon = 0.95$ | 00:00 | 06:00 | 12:00 | 18:00 |
|---|---|---|---|---|
| *00:00* | | z = 0.14 <br> p > 0.5 | *z = 10.48* <br> *p < 0.5* | z = 0.45 <br> p > 0.5 |
| *06:00* | z = 0.14 <br> p > 0.05 | | *z = 10.70* <br> *p < 0.05* | z = 0.34 <br> p > 0.05 |
| *12:00* | **z = 10.48** <br> **p < 0.05** | **z = 10.70** <br> **p < 0.05** | | **z = 10.85** <br> **p < 0.05** |
| *18:00* | z = 0.45 <br> p > 0.05 | z = 0.34 <br> p > 0.05 | *z = 10.85* <br> *p < 0.05* | |

(b)

| $\epsilon = 0.99$ | 00:00 | 06:00 | 12:00 | 18:00 |
|---|---|---|---|---|
| *00:00* | | z = 0.14 <br> p > 0.5 | *z = 10.46* <br> *p < 0.5* | z = 0.45 <br> p > 0.5 |
| *06:00* | z = 0.14 <br> p > 0.05 | | *z = 10.69* <br> *p < 0.05* | z = 0.34 <br> p > 0.05 |
| *12:00* | **z = 10.46** <br> **p < 0.05** | **z = 10.69** <br> **p < 0.05** | | **z = 10.84** <br> **p < 0.05** |
| *18:00* | z = 0.45 <br> p > 0.05 | z = 0.34 <br> p > 0.05 | *z = 10.84* <br> *p < 0.05* | |

## 4.4 Near-surface temperature inversions

The similarity in MODIS brightness and surface temperature offsets from air temperatures in fall and winter may be also related to more frequent near-surface temperature inversions during those months. Results showing that the MODIS LST offset is highly correlated with the level of solar radiation supports the hypothesis that a near-surface temperature inversion is the primary driver of the observed offset. Incoming solar radiation is lowest in the fall and winter, when the offset is greatest, and therefore may be a root control on the seasonal nature of the cold offset. Low wind speeds maintain existing near-surface inversions; however, solar radiation is the primary control on inversion development, providing an explanation for the weaker correlation between the LST offset and wind speed. Observed wintertime MODIS LSTs show a median offset of greater than 8°C at both Divide and Eclipse (Table 2). Results from the simple energy balance model support these observations,

predicting a median wintertime temperature inversion of 4.8°C ($\epsilon_s = 0.95$) and 7.4°C ($\epsilon_s = 0.99$). However, wintertime near-surface temperature inversions have been observed at other glaciated sites (where both air and surface temperatures have been measured in situ), but with smaller magnitudes than the MODIS LST offset and predicted inversions at Divide and Eclipse. Surface temperatures at the South Pole during the winter of 2001 were a median of 1.3°C lower than 2 m air temperatures under clear sky conditions (Hudson and Brandt, 2005). Likewise, surface temperatures at Summit, Greenland were $1.5 \pm 0.2$°C lower than 2 m air temperatures during the winter of 2008–2009 (Koenig and Hall, 2010). The smaller magnitude of surface-air temperature offsets at Summit, Greenland and the South Pole relative to our study sites may be due to a stronger influence of turbulent fluxes at Summit, Greenland and the South Pole or to variations in albedo, as both turbulent fluxes and surface albedo can be strong controls on surface energy balance (Braithwaite and Olesen, 1990; Oerlemans, 1991; Ebrahimi and Marshall, 2016).

In comparing the magnitude of the summer LST offset here (JJA Mdn $= -1.0$°C), to prior studies, the offsets presented here are smaller than previously observed summer MODIS LST offsets in the St. Elias (5–7°C, Williamson et al. 2017). However, these prior LSTs were daily averages of maximum and minimum values, with most of the offset being attributed to the inclusion of minimum LSTs (Williamson et al., 2017). In contrast, this study uses a single daily LST value and coincident AWS measurements acquired between 12:00 and 13:00 (local solar time), when surface and air temperatures are near their maximum, thereby eliminating the effects of any diurnal cycle on observed LST offsets. Our modeled temperature inversions show a diurnal cycle, which is more dramatic in the summer than the winter because of the greater difference between incoming solar radiation during the day and night, and is likely responsible for the higher magnitude of the previously observed summer LST offsets (Fig. 11; Tables 6, 7). The magnitude of the summer LST offset at Eclipse and Divide is in closer agreement with temperature inversions observed at Summit, Greenland, where 2 m air and surface temperatures have been contemporaneously measured in situ. During June–July 2015, Summit, Greenland surface temperatures were 0.32 to 2.4°C lower than 2 m air temperatures (Adolph et al., 2018). At three northern Alaska sites, summer clear-sky surface temperatures were higher than corresponding 2 m air temperatures (Barrow and Atqasuk in 2010, and Olitok Point in 2014; (Good, 2016). In contrast to sites in Greenland and the St. Elias, these northern Alaskan sites are characterized by seasonal snow cover. Sites with seasonal snow cover present challenges for interpretation because they experience surface melt and a drastic change in surface type over the course of the melt season. Across glaciated areas, sites in the accumulation zone have been found to have the weakest near-surface inversions during the summer, while sites in the ablation zone have been found to have the strongest near-surface inversions during the summer, likely because of the change in surface type with over the melt season (Nielsen-Englyst et al., 2019).

Results from the simple energy balance model predict no summertime inversion at all, with surface temperatures being a median of 0.8°C higher than 2 m air temperatures. The dip in modeled summer temperature inversions (Fig. 10) is the result of our 0°C surface temperature cap, which is a simplistic numerical correction for unrealistically high summer surface temperatures over 0°C. Because the 0°C cap is applied after the calculation of surface temperatures and does not address the mechanisms of inversion development, the distinction between capped temperatures slightly over 0°C and uncapped temperatures slightly

below 0°C is somewhat arbitrary. We therefore focus on the magnitudes and seasonal patterns of calculated inversions during summer and winter rather than during the shoulder seasons where the temperature cap likely biases our results.

Discrepancies between modeled temperature inversions and observed LST offsets likely arise from variations in albedo, which has a strong control on surface energy balance (Oerlemans, 1991; Ebrahimi and Marshall, 2016). We use an albedo value of $\alpha = 0.742$, but albedo values from $\alpha = 0.661$–$0.831$ have been measured at Divide (Williamson et al., 2016). Using an albedo of $\alpha = 0.661$ and an emissivity of $\epsilon_s = 0.95$, modeled summer surface temperatures are a median of 38.7°C higher than 2 m air temperatures prior to applying the 0°C surface temperature cap. Modeled winter surface temperatures are a

median of 2.2°C lower than 2 m air temperatures. Using an albedo of $\alpha = 0.831$ and an emissivity of $\epsilon_s = 0.95$, modeled summer surface temperatures are a median of 17.2°C higher than 2 m air temperatures, and winter surface temperatures are a median of 8.4°C lower than 2 m air temperatures. Our albedo value of $\alpha = 0.742$, measured in August when the snow can be relatively dirty, may be an underestimate during parts of the year when debris is more limited.

     Additionally, we do not take turbulent fluxes into account in modeled surface temperatures. Turbulent fluxes serve to dis-

435 mantle inversions, so we interpret modeled temperature differences to represent an upper bound of expected inversion strength. Overall, the uncertainty in albedo and omission of turbulent fluxes in our modeling lead to wide uncertainty in calculated surface temperatures and inversion strength. However, our simplistic approach is sufficient to explore the physical plausibility of near-surface temperature inversions in the St. Elias. Results suggest that near-surface inversions are plausible at Divide and Eclipse and may account for most of the observed offset in MODIS LSTs. To our knowledge, results here provide the first

evidence for near-surface temperature inversions in a heterogeneous alpine environment, as well as the first exploration of their seasonal and diurnal signals in such an environment. We recommend continued work to understand near-surface thermal processes in these complex regions, including obtaining in situ air and surface temperatures to validate these results.

### 4.5   Approximating air temperatures from MODIS LSTs

Despite some uncertainty about the exact mechanism for the MODIS offset, and the lack of a conversion to air temperatures

based on physical principles, MODIS LSTs can still be used with sparse air temperature measurements to shed light on the relationship between climatic conditions and surface processes on interannual timescales. In particular, the MODIS LSTs may allow closer examination of surface melt and mass balance in the North Pacific, as the offset is relatively minor during the summer melt season (Table 2). Surface melt correlates with air temperatures, largely because of increased longwave atmospheric radiation (an important source of energy for melt) with higher temperatures (Ohmura, 2001; Cuffey and Paterson, 2010).

Higher air temperatures tend to occur under cloudy conditions when no MODIS imagery is available (Walsh and Chapman, 1998). MODIS LSTs may therefore be inadequate for examining temperature conditions associated with individual extreme melt events. However, MODIS LSTs still have utility when examining melt on interannual timescales.

     Various methods have been used to convert MODIS LSTs to air temperatures, including advanced statistical and modeling frameworks (e.g. Hengl et al., 2012; Benali et al., 2012; Emamifar et al., 2013; Zhu et al., 2013; Janatian et al., 2017; Zhang

et al., 2016; Hooker et al., 2018; Zhang et al., 2018; Zhang et al., 2021). Our simple linear regression effectively converts

MODIS LSTs to air temperatures, enabling their use for both qualitative and quantitative applications related to glacier melt and mass balance on annual timescales.

We recommend converting mean annual MODIS LSTs to air temperatures and using these in conjunction with regional glacier mass balance data to track current temperature changes and glacier response on a broad scale. We also recommend using converted mean annual MODIS LSTs in the interpretation of refrozen melt archived in ice cores drilled at sites without long in situ temperature records. Qualitatively, MODIS LSTs (converted or unconverted) can be used to evaluate whether years of high surface temperatures correspond to years of high amounts of melt in the ice core record. If they do, converted LSTs can be used to quantitatively describe the relationship between air temperature and archived melt, enabling the use of refrozen melt as a temperature proxy.

## 5 Conclusions

Remote sensing is a powerful tool to obtain information about surface conditions at inaccessible locations; however, oftentimes these measurements need calibration and validation. Here we investigated an observed offset in MODIS LSTs from AWS air temperatures in the St. Elias Mountains (Yukon, Canada), and found the offset to be most pronounced in the fall and winter. We tested three hypotheses for the origin of the offset: (a) the large spatial footprint of the MODIS sensor in highly heterogeneous alpine terrain, (b) poorly constrained snow emissivity values, and (c) a real temperature difference between the surface and air due to near-surface temperature inversions. We found that the MODIS sensor's large footprint does not account for the offset in its LSTs. Even in highly heterogeneous alpine terrain, the spatial coherence of temperatures across study sites in the region makes it doubtful that offsets from AWS temperatures in excess of 10°C could be regularly obtained by averaging temperature across a single square kilometer to produce the MODIS LST. Moreover, surface temperatures from the ASTER sensor, which has a footprint of 90 m as compared to MODIS' 1 km footprint, still exhibit an offset relative to AWS measurements. The MODIS LST offset is therefore not simply an error arising from the spatial resolution of MODIS data. We also found that poorly constrained snow emissivity values fail to account for the MODIS LST offset; a pronounced fall and winter offset between MODIS brightness temperatures and AWS temperatures is present even prior to the incorporation of snow surface emissivity. However, poorly constrained fall and winter snow emissivity values may exacerbate an existing offset, particularly after snowfall events, when emissivity is likely to change rapidly due to settling and compaction processes. In short, emissivity values are not responsible for the production of the MODIS LST offset, but their role in amplifying it remains unknown.

We found that the physical conditions (low wind speeds, low levels of incoming solar radiation) associated with greater MODIS LST offsets at Eclipse and Divide are consistent with near-surface temperature inversions measured over Greenland (Adolph et al., 2018). In modeling near-surface temperature inversions, we found observed MODIS LST offsets to be within the range of expected inversions based on Divide AWS and ERA5 reanalysis data, supporting the hypothesis that the MODIS LST offset is representative of a physical difference between the properties measured by MODIS (surface temperature) and weather stations (air temperature) rather than the instrumentation or algorithm used to calculate LSTs. Our results provide, to our knowledge, the first evidence for near-surface temperature inversions in a heterogenous alpine environment. Although

results do not preclude errors in the MODIS sensor or the LST algorithm, they indicate that near-surface inversions require consideration when estimating the surface energy balance of rapidly changing glaciated alpine regions.

Finally, we show that interannual patterns in MODIS LSTs are in good agreement with those of AWS temperature measurements in an alpine environment at Eclipse and Divide. On annual timeframes, we were able to convert MODIS LSTs to air temperatures consistent with AWS measurements by applying a linear conversion $T_{air,approx.} = -3.73 + 0.44 LST$. While winter and fall LST offsets remain larger than those in spring and summer, the established conversion factor enables a more accurate assessment of melt conditions year to year in alpine environments. This work provides a step forward in using remote sensing imagery to expand in situ records and thus provide insight into past and present temperature changes in the St. Elias Mountains and broader North Pacific region.

*Data availability.* MODIS and Landsat data are freely available from the U.S. Geological Survey (https://lpdaacsvc.cr.usgs.gov/appeears/). Likewise, ASTER data are freely available from NASA (https://search.earthdata.nasa.gov/search). Eclipse iButton data are available upon request from Karl Kreutz. Eclipse and Divide AWS data are available upon request from Luke Copland.

*Author contributions.* Ingalise Kindstedt, Kristin M. Schild, and Karl Kreutz formulated the research goals, hypotheses and testing methods. Karl Kreutz, Dominic Winski, Seth Campbell, Luke Copland, and Erin McConnell participated in fieldwork and data collection. Ingalise Kindstedt completed data analysis with contribution from Kristin M. Schild and Dominic Winski. Ingalise Kindstedt prepared the manuscript with contributions from all co-authors.

*Competing interests.* The authors declare that they have no conflict of interest

*Acknowledgements.* We thank the National Science Foundation awards #2002483 and #1502783, the National Aeronautics and Space Administration, the Maine Space Grant Consortium, the University of Maine Climate Change Institute, the Canada Foundation for Innovation, Polar Continental Shelf Program, and the Natural Sciences and Engineering Research Council of Canada for support for this work. We thank Icefield Discovery for logistical support, Gerald Holdsworth and Scott Williamson for providing the Eclipse Icefield AWS data, and Christian Zdanowicz and Dorothy K. Hall for additional contributions to data collection.

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
