# Peer review of "Offset of MODIS land surface temperatures from in situ air temperatures in the Upper Kaskawulsh Glacier region (St. Elias mountains) indicates near-surface temperature inversions"

_The Cryosphere, 2021_

## Referee Comment (RC1)

Review of "Evaluating sources of an apparent cold bias in MODIS land surface temperatures in the St. Elias Mountains, Yukon, Canada"

**Overview:**

This manuscript describes analysis of in-situ air temperature data relative to the MODIS LST product (MYD21) with supplemental data from ASTER, Landsat, and additional meteorological measurements at two sites in the St. Elias Mountains. The central focus of the manuscript is to determine the causes of difference between in-situ air temperature measurements and MODIS LST data (called the "MODIS offset"). Three main hypotheses are explored: 1) The large footprint of the MODIS pixels results in the offset; 2) The lack of constraint in surface emissivity results in the offset; 3) Near-surface temperature inversions lead to this offset. The work demonstrates that the first two hypothesis are unlikely to play a large role in the offset, and that the near-surface inversions are likely a reason why there is a difference between in-situ air temperature and MODIS LSTs. While this has been shown in previous studies, the authors indicate that this is the first time that inversions have been identified as a cause of this offset in this type of environment. The manuscript makes a compelling case that the system studied is an important one given the role of Alaskan glaciers in sea level rise and the fact that the surface temperatures in this region are understudied. Understanding how representative MODIS LSTs (and other remote sensing surface temperature products) are of actual surface conditions is critical to effectively monitoring this region.

Overall, this is important work and a useful contribution. I think that the work can be improved by increased clarity and discussion about the fact that a comparison of 2m air temperature and MODIS LST is not a 1:1 comparison. I see that it is addressed in the text, but I believe it needs to be a bit more explicit. For example: in Section 4.4, I think the language of "corrected" MODIS LST is a bit misleading because there isn't a true error in MODIS LSTs identified; it is simply determined that it is not equivalent to 2m air temperature. The linear regression implemented is a tool to adjust the LST to an air temperature. Furthermore, in the abstract and conclusion section, I think it is important to clarify that this analysis doesn't rule out the possibility that there may be measurement errors in the MODIS algorithm for determining surface temperature because there was not a direct comparison to in-situ skin temperature in this study. While convincing evidence is provided that inversions are likely a source of difference between MODIS LST and in-situ 2m air temp, it is still possible that MODIS LSTs have error relative to in-situ skin temp too. Lastly, in the title, I would be hesitant to call this a cold bias without explicitly stating that the "bias" is relative to air temperature measurements.

I have several specific comments to improve clarity and increase detail about methodology, but the manuscript is overall well-written, the figures are clear, and the methods are sound. This is a very rich dataset, and the analysis and results provide an important contribution to our understanding of near surface temperatures in understudied snow and ice covered regions.

**Specific Comments:**

Lines 4-5: In the statement that MODIS LSTs are offset from AWS data, is this referring to prior studies or the current work? Please clarify.

Lines 15-17: I agree that understanding near-surface physical properties is critical to convert MODIS LSTs from a surface temperature measurement to an air temperature measurement, but this is different

from improving the accuracy of the MODIS LST, which this work doesn't directly address because it compares in-situ air temp (and not in-situ skin temp) to MODIS LSTs. Please revise this statement accordingly.

Line 41: The statement that all in-situ temperatures are air temperatures is key here. I think it requires a bit of further discussion. Given the Guillevic et al. (2017) report, validation should be done with skin temperature wherever possible. I understand that in-situ skin temperature data is not available in this case, and I believe comparison to in-situ air temperature is a worthy endeavor, but I think it should be explicitly addressed here (or elsewhere) that this is different from a standard validation of the MODIS LST product.

Table 1: Consider adding to this table further information about the footprint of each sensor, the specific products/instruments used, and the uncertainty associated with each measurement (if available).

Lines 118-122: In the window when the two sensors at Divide overlap, which dataset is used?

Lines 123-125: Was the container containing the iButton sensor ventilated? Was it a light color to limit absorbed solar radiation? I know that there can be issues with iButton sensors heating up during periods of high incoming solar radiation.

Lines 125-126: It looks in Figure 3 like the Eclipse Weather Station and Eclipse iButton datasets do not overlap, so it's not clear to me how it was determined that the records were consistent. Please clarify and provide data if needed, perhaps in a supplement.

Lines 135-143: I think that choosing to look at a nearby MODIS pixel that does not include the darker nunatak surfaces is probably a good idea, but the implications of this choice should be further explored. Perhaps the air temperature above the dark surface actually is higher than the air temperature above the nearby ice/snow. Comparing the MODIS pixels and providing that information in Table 2 is great, and I think that in the discussion of results, the manuscript should come back to this and address what the implications of this choice are on the results.

Line 145: I see that later in the manuscript there is a justification provided for choosing to only consider data from the time window from 11:00 – 1:30. I think it would be appropriate to address this here in the methods.

Lines 145-147: Just to clarify, the 742 images at Divide span 20 years of data, and the 100 images at Eclipse span ~2 years of data. Is that correct? Please state that clearly here. Also, please specify how the MODIS images and AWS data are paired in time. Is it simply the closest hourly measurement that is paired with the MODIS temperature, or is it some kind of average of multiple AWS measurements?

Section 2.3: Please clarify if only ASTER and MODIS images from the 11-1:30 window that had paired AWS data are used in this comparison.

Lines 168-169: Please explain how Landsat images are "examined for cloud cover." Is there a particular algorithm used, and are particular thresholds implemented?

Equations 2+3: I believe it might be appropriate to use ≈ instead of = in these expressions.

Line 200: Provide a bit more information about the atmospheric emissivity from ERA5.

Line 202: Please provide a bit of description of the surface types that are present at the two sites. Is surface melt common? This doesn't necessarily need to be right here, but the point about albedo raised the question of if this is a reasonable assumption or not.

Section 3.1 (and elsewhere): When comparing different seasons, I think that the results of the Wilcoxon rank sum tests can be referenced in the tables and don't need to be repeated in each paragraph. The many parentheticals make it a bit challenging to read and the statistics are clearly presented in the tables. Maybe the journal or author preferences dictate that it should also be in each paragraph, but I think it interferes a bit with clarity.

Figure 4: 1:1 plots of in-situ air temperature vs. MODIS LST are very instructive, and should potentially be included in addition to these box plots.

Lines 247-248: Please indicate if it was tested that the data are normally distributed prior to linear regression, and normalize data if necessary.

Figure 8: Adding a horizontal line at 0 C would be helpful in figures a and b; Is this data for all 20 years when Divide AWS data are available?

Figure 9: Adding a horizontal line at 0 C would be helpful in figures a and b

Lines 287-290: Is it possible that the lack of difference between offsets in MODIS brightness and surface temperatures in the fall and winter is also because of increased inversions relative to other seasons?

Lines 294-295: I would potentially expect a wider variability in surface conditions (and thus emissivity) in the summer, but this may be site specific. Perhaps explain more why winter could result in more rapidly changing emissivity.

Lines 304-305: The analyses exploring links between accumulation and the LST offset seem to have covered a wide array of options. Was "days since last accumulation" one of the ways this data was analyzed? I couldn't quite tell from the descriptions and that might be a reasonable metric to consider.

Lines 314-315: Why used different emissivity values for the different sites, or is this a typo?

Line 321: Is it a discrepancy because it is not accounted for given the assumptions used in the T_surface calculations presented here?

Lines 323-338: Consider also comparing to 2m near surface inversions investigated in Nielsen-Englyst et al. (2019).

Line 342-343: Perhaps I'm missing something, but why wouldn't winter surface temperatures also be high then?

Lines 370-375: Please explain in a bit more detail how the data are processed before going into the linear regression. I suggest framing the linear fit with the actual variables desired (instead of generic x and y). The corrected MODIS temperatures have a low mean error because of how they are calculated, but how does the standard deviation compare in magnitude to the interannual variability? Is a better fit to the data achieved if it is not averaged annually but instead each paired data point (of 2 m air temp and MODIS LST) is part of the regression?

**Minor Comments:**

Line 42: Consider starting a new paragraph with the sentence that begins "Instrumentation…"

Line 68: Consider phrasing as "apparent MODIS LST offset" here and elsewhere?

Line 86: Consider adding a definition of a nunatak

Lines 88-89: Consider reversing sentence that begins with "Here…" so that it discusses inversions first, which were just being discussed. I found the sentence a bit confusing on the first read because I was unsure which hypotheses were being referenced.

Line 221 (and elsewhere): forgotten degrees C units on reported median.

Line 222: "MODIS temperature difference data spaces >10 degrees C" – is this saying the maximum offset is greater than 10 degrees C? Please clarify.

Line 249: Perhaps an "overwhelming majority" or some other language would be better suited to describe 95%.

Line 256: "modeled temperature difference data is greater than 60 degrees C" – does this mean that the modeled inversions are sometimes up to 60 degrees? Please clarify.

Lines 275-280: Since the manuscript already revealed that the brightness temp and surface temp do show the same pattern, going through the hypothetical scenarios in the way they are framed confused me a bit. Consider rephrasing to make it clear that one of them we already know to be what was observed.

**Citations:**

Guillevic, P., Göttsche, F., Nickeson, J., Hulley, G., Ghent, D., Yu, Y., Trigo, I., Hook, S., Sobrino, J.A., Remedios, J.,Román, M. & Camacho, F. (2017). Land Surface Temperature Product Validation Best Practice Protocol. Version 1.0. In P. Guillevic, F. Göttsche, J. Nickeson & M. Román (Eds.), Best Practice for Satellite-Derived Land Product Validation (p. 60): Land Product Validation Subgroup (WGCV/CEOS), doi:10.5067/doc/ceoswgcv/lpv/lst.001

Nielsen-Englyst, P., Høyer, J. L., Madsen, K. S., Tonboe, R., Dybkjær, G., & Alerskans, E. (2019). In situ observed relationships between snow and ice surface skin temperatures and 2 m air temperatures in the Arctic. The Cryosphere, 13(3), 1005-1024.

---

## Author Comment (AC1)

Thank you for your constructive comments. We have addressed your general comments in a bulk response and your specific comments line by line below. Comments are indicated by boldface and italicized text; our responses in normal text and preceded by [Author response].

*Review of "Evaluating sources of an apparent cold bias in MODIS land surface temperatures in the St. Elias Mountains, Yukon, Canada"*

*Overview:*

*This manuscript describes analysis of in-situ air temperature data relative to the MODIS LST product (MYD21) with supplemental data from ASTER, Landsat, and additional meteorological measurements at two sites in the St. Elias Mountains. The central focus of the manuscript is to determine the causes of difference between in-situ air temperature measurements and MODIS LST data (called the "MODIS offset"). Three main hypotheses are explored: 1) The large footprint of the MODIS pixels results in the offset; 2) The lack of constraint in surface emissivity results in the offset; 3) Near-surface temperature inversions lead to this offset. The work demonstrates that the first two hypothesis are unlikely to play a large role in the offset, and that the near-surface inversions are likely a reason why there is a difference between in-situ air temperature and MODIS LSTs. While this has been shown in previous studies, the authors indicate that this is the first time that inversions have been identified as a cause of this offset in this type of environment. The manuscript makes a compelling case that the system studied is an important one given the role of Alaskan glaciers in sea level rise and the fact that the surface temperatures in this region are understudied. Understanding how representative MODIS LSTs (and other remote sensing surface temperature products) are of actual surface conditions is critical to effectively monitoring this region.*

*Overall, this is important work and a useful contribution. I think that the work can be improved by increased clarity and discussion about the fact that a comparison of 2m air temperature and MODIS LST is not a 1:1 comparison. I see that it is addressed in the text, but I believe it needs to be a bit more explicit. For example: in Section 4.4, I think the language of "corrected" MODIS LST is a bit misleading because there isn't a true error in MODIS LSTs identified; it is simply determined that it is not equivalent to 2m air temperature. The linear regression implemented is a tool to adjust the LST to an air temperature. Furthermore, in the abstract and conclusion section, I think it is important to clarify that this analysis doesn't rule out the possibility that there may be measurement errors in the MODIS algorithm for determining surface temperature because there was not a direct comparison to in-situ skin temperature in this study. While convincing evidence is provided that inversions are likely a source of difference between MODIS LST and in-situ 2m air temp, it is still possible that MODIS LSTs have error relative to in-situ skin temp too. Lastly, in the title, I would be hesitant to call this a cold bias without explicitly stating that the "bias" is relative to air temperature measurements.*

*I have several specific comments to improve clarity and increase detail about methodology, but the manuscript is overall well-written, the figures are clear, and the methods are sound. This is a very rich dataset, and the analysis and results provide an important contribution to our understanding of near surface temperatures in understudied snow and ice covered regions.*

[Author response]
We understand that clarity regarding the difference between 2 m air temperature and MODIS LST is essential, and we will address this more explicitly. For example, see our edited statement in response to comments on Line 41. We will also add statements to the abstract and conclusion clarifying that this study does not rule out errors in the MODIS algorithm. Finally, we will change our title to "Offset of MODIS land surface temperatures from in situ measurements in the Upper Kaskawulsh Glacier region (St. Elias

mountains) indicates near-surface temperature inversions" and insert the following statement preceding our goal to clarify the distinction between air and surface temperatures.

Edited Line 99: Lastly, it may be that the LST offset does not arise during the calculation of LSTs at all, but is a real physical temperature difference between the surface and air due to the development of a near-surface temperature inversion. Although MODIS LSTs can be a useful complement to in situ air temperatures, the two cannot be directly compared and physical differences between the two must be accounted for when using them together.

*Specific Comments:*

*Lines 4-5: In the statement that MODIS LSTs are offset from AWS data, is this referring to prior studies or the current work? Please clarify.*
[Author response]
This statement is in reference to prior studies.
Updated text to read: However, MODIS LSTs in the St. Elias Mountains have been found in prior studies to show an offset from available weather station measurements, the source of which is unknown.

*Lines 15-17: I agree that understanding near-surface physical properties is critical to convert MODIS LSTs from a surface temperature measurement to an air temperature measurement, but this is different from improving the accuracy of the MODIS LST, which this work doesn't directly address because it compares in-situ air temp (and not in-situ skin temp) to MODIS LSTs. Please revise this statement accordingly.*
[Author response]
Updated text to read: These results demonstrate that efforts to derive an air temperature measurement from MODIS LST data should focus on understanding near-surface physical properties rather than refining the MODIS sensor or LST algorithm.

*Line 41: The statement that all in-situ temperatures are air temperatures is key here. I think it requires a bit of further discussion. Given the Guillevic et al. (2017) report, validation should be done with skin temperature wherever possible. I understand that in-situ skin temperature data is not available in this case, and I believe comparison to in-situ air temperature is a worthy endeavor, but I think it should be explicitly addressed here (or elsewhere) that this is different from a standard validation of the MODIS LST product.*
[Author response]
Updated text to read: Remote sensing temperatures include the final surface temperature product, as well as "brightness temperature", or the surface temperature of a perfect blackbody emitter under the same conditions. In contrast, temperatures measured in situ are directly measured using instruments onsite, and can be measured for both the earth's surface and the air above it. Surface temperatures measured in situ provide important validation for remote sensing surface temperatures such as MODIS LSTs. However, because of a lack of in situ surface temperature data in our study region, unless otherwise stated, all in situ temperatures used here refer to the air ~2m above the land surface. Our study is therefore not a standard validation of the MODIS LST product, but rather an evaluation of its use in conjunction with in situ air temperatures to characterize the near-surface temperature conditions of the St. Elias region.

*Table 1: Consider adding to this table further information about the footprint of each sensor, the specific products/instruments used, and the uncertainty associated with each measurement (if*

*available).*

[Author response]
We will update Table 1 to contain the below information:

| | Name | Measurement technique | Footprint | Product/instrument | Uncertainty |
|---|---|---|---|---|---|
| Air temperatures | Divide AWS | In situ | Point measurement | Campbell 107F | ± 0.2°C |
| | | | | HOBO S-THB-M008 12-bit sensor | ± 0.21°C |
| | Eclipse AWS | In situ | Point measurement | | |
| | iButton | In situ | Point measurement | Maxim Integrated iButton Data Logger DS1922L | ± 0.5°C |
| Surface temperatures | MODIS LST | Remote sensing | 1 km | MYD21 | |
| | ASTER surface temperature | Remote sensing | 90 m | AST_08 | |
| | MODIS BT | Remote sensing | 1 km | MODTBGA_006 | |
| | ASTER BT | Remote sensing | 90 m | Calculated from ASTL1T following Ndossi and Avdan (2016) | |
| | Landsat BT | Remote sensing | Resampled from 100 m to 30 m (Landsat 8) | LC08 | |
| | | | Resampled from 120 m to 30 m (Landsat 5,7) | LE07 LT05 | |

*Lines 118-122: In the window when the two sensors at Divide overlap, which dataset is used?*
In the window where the two sensors at Divide overlap, the HOBO S-THB-M008 12-bit sensor was used because it provides contemporary solar radiation, relative humidity, wind speed, and pressure data.

*Lines 123-125: Was the container containing the iButton sensor ventilated? Was it a light color to limit absorbed solar radiation? I know that there can be issues with iButton sensors heating up during periods of high incoming solar radiation.*
[Author response]
Available temperature data at Eclipse are lower quality than at Divide, with limited temporal coverage and sensors not up to World Meteorological Organization standards. We therefore focus on data from Divide, but include available data from Eclipse with the caveat that results are less robust. Temperatures at Eclipse were obtained from an AWS from 2005-2007, and a Maxim Integrated iButton Data Logger DS1922L (±0.5°C) from 21 May 2016 to 17 May 2017, both located on or near a bedrock outcrop ~3 km from the site of an ice core drilled at Eclipse in 2016 (Fig. 3). The AWS recorded hourly averages of 5 minute sampling intervals using digital sensors housed in a passively vented radiation shield at a height of approximately 2 m (Williamson et al., 2020). The iButton recorded temperatures at 3-hour

intervals and was placed inside an unvented clear plastic container shielded with rocks. Because data is so limited at Eclipse, we combine the AWS and iButton datasets for maximum coverage at the site. We refer to both the Divide AWS and the combined Eclipse iButton and AWS data as "AWS" for the remainder of this paper.

***Lines 125-126: It looks in Figure 3 like the Eclipse Weather Station and Eclipse iButton datasets do not overlap, so it's not clear to me how it was determined that the records were consistent. Please clarify and provide data if needed, perhaps in a supplement.***
[Author response]

We did not perform a robust test of the datasets' consistency, instead choosing to focus on data from Divide, which is more abundant. Around 88% of the temperature data used in this study came from Divide. Additionally our examination of other meteorological variables and our surface energy balance calculations are all performed with data from Divide. However, we still included what data we had from Eclipse to supplement the results at Divide. See edited statement above in response to comment on Lines 123-125.

***Lines 135-143: I think that choosing to look at a nearby MODIS pixel that does not include the darker nunatak surfaces is probably a good idea, but the implications of this choice should be further explored. Perhaps the air temperature above the dark surface actually is higher than the air temperature above the nearby ice/snow. Comparing the MODIS pixels and providing that information in Table 2 is great, and I think that in the discussion of results, the manuscript should come back to this and address what the implications of this choice are on the results.***
[Author response]
Updated text to read: Our goal is to determine the dominant source of the offset in MODIS LSTs at glaciated sites in the St. Elias. Because the Eclipse and Divide AWS are located on nunataks, we test for the LST offset using MODIS data encompassing adjacent ice core sites ~3 km from each AWS location, thereby excluding the dark nunatak surface from the MODIS pixel and focusing on the ice surface (Fig. 2). We compute the difference in MODIS LST between the ice core site grid cell (containing only ice) and the AWS site grid cell (containing ice and rock) to determine whether the inclusion of the nunatak has a discernible effect on the MODIS LST.

To be inserted in results section (and add figure plotting temperature differences between AWS and ice core sites): MODIS data at the Divide AWS nunatak and adjacent ice core site have a median temperature difference of 0.86°C and interquartile range of 1.97°C. The difference between the two sites shows greater variability in the fall (IQR = 3.21) and winter (IQR = 3.98) than in the spring (IQR = 0.70) and summer (IQR = 1.80), with the ice core site tending to be slightly colder in the winter (median temperature difference of -0.49°C), but warmer in the spring (median = 0.95°C), summer (median = 1.33°C) and fall (median = 0.28°C). Temperature differences between the Divide AWS and ice core site are summarized in Table 2.

Note: Table 2 will be edited to show differences between the ice core and AWS sites as medians rather than means so as to be more directly comparable with median LST offsets from AWS temperatures.

To be inserted in the discussion section: MODIS LSTs at the ice core site do not tend to be colder than at the AWS site except during the winter. The inclusion of the warmer nunatak surface in the MODIS grid cell at the AWS site fails to provide a compelling explanation for the colder wintertime LSTs at the ice core site, given that more of the rock surface would likely have snow cover during the winter. The colder wintertime

LSTs at the ice core site may contribute to the MODIS LST offset from in situ temperature measurements examined in this study. However, this contribution is too small (median = -0.49°C) to explain the magnitude of the MODIS LST offset at the Divide ice core site (median = -8.40°C). In the spring, summer, and fall, the LSTs at the ice core site tend to be slightly warmer than at the AWS site. Results here may therefore underestimate the magnitude of the MODIS LST offset from AWS temperatures in these seasons.

***Line 145: I see that later in the manuscript there is a justification provided for choosing to only consider data from the time window from 11:00 – 1:30. I think it would be appropriate to address this here in the methods.***
**[Author response]**
The comment provided in the results section (lines 326-328) serves to point out that we don't capture diurnal cycles in our data because of the limited time period of data acquisition; however this is a result of, rather than a reason for, limiting data acquisition to the period from 12:00-13:00. MODIS overpass times are all from approximately 11:00-13:30. We further narrow our data acquisition to the hours for which the viewing angle of our site is < 30°, which are from 12:00-13:00.
Updated text to read: Temperature differences between the Divide AWS and ice core site are summarized in Table 2. MODIS LST data were obtained for the period 2000-2020 (https://lpdaacsvc.cr.usgs.gov/appeears/) for dates with minimal cloud cover between the hours of 12:00 and 13:00 (local solar time), when viewing angle is less than 30°, to mitigate the effect of viewing angle on temperature and emissivity. At Divide, 742 MODIS images spanning 2002-2020 were analyzed. Seasonally, 203 images were acquired in spring (MAM), 169 in summer (JJA), 188 in fall (SON), and 182 in winter (DJF). The average time between scenes at Divide was 9 days after filtering. At Eclipse, 100 MODIS images were analyzed: 87 spanning June 2005 through June 2007 and 13 spanning November 2016 through February 1017. Each MODIS image was paired with the closest hourly measurement available in the AWS data.

***Lines 145-147: Just to clarify, the 742 images at Divide span 20 years of data, and the 100 images at Eclipse span ~2 years of data. Is that correct? Please state that clearly here. Also, please specify how the MODIS images and AWS data are paired in time. Is it simply the closest hourly measurement that is paired with the MODIS temperature, or is it some kind of average of multiple AWS measurements?***
**[Author response]**
Addressed above

***Section 2.3: Please clarify if only ASTER and MODIS images from the 11-1:30 window that had paired AWS data are used in this comparison.***
**[Author response]**
Updated text to read: To test if the LST offset is a result of the MODIS sensor's large footprint, we calculate the offset of both ASTER (90 m footprint) and MODIS (1 km footprint) surface temperatures from AWS measurements and then compare the magnitude of the offsets. We use only ASTER and MODIS images from the 12:00-13:00 window that had paired AWS data in this comparison.

***Lines 168-169: Please explain how Landsat images are "examined for cloud cover." Is there a particular algorithm used, and are particular thresholds implemented?***
**[Author response]**
Landsat imagery was visually examined for cloud cover. Only images with no visibly identifiable clouds were analyzed.
Updated text to read: Landsat top of atmosphere brightness temperature imagery

(https://earthexplorer.usgs.gov/) was visually examined for cloud cover, and cloud-free pixels were extracted for analysis using QGIS.

*Equations 2+3: I believe it might be appropriate to use ≈ instead of = in these expressions.*
**[Author response]**
Updated text to read:

$$E_N \approx E_S\downarrow + E_S\uparrow + E_L\downarrow + E_L\uparrow$$

$$T_S \approx \left(\frac{E_L\downarrow + E_S\downarrow(1-\alpha)}{\epsilon_S\sigma}\right)^{0.25}$$

*Line 200: Provide a bit more information about the atmospheric emissivity from ERA5.*
**[Author response]**
Updated text to read: We calculate downward longwave radiation as follows, using 2 m air temperature ($T_a$) from Divide and atmospheric emissivity ($\epsilon_a$) derived from the ERA5 reanalysis longwave radiation product. We use only the derived emissivity from the ERA5 product, rather than the total downward radiation in order to use measured values (in situ 2 m air temperature) where possible. ERA5 outputs have a spatial resolution of 31 km; data span 2002-2019 every six hours (Hersbach et al., 2020). Atmospheric emissivity increases with increasing surface vapor pressure (Staley and Jurica, 1971). Our atmospheric emissivity values ranged from ~0.48 to 1. Atmospheric emissivity measured over the Sierra Nevada (Spain) from 2005-2011 ranged from ~0.4-1 (Herrero and Polo, 2012).

*Line 202: Please provide a bit of description of the surface types that are present at the two sites. Is surface melt common? This doesn't necessarily need to be right here, but the point about albedo raised the question of if this is a reasonable assumption or not.*
**[Author response]**
Surface melt is present but limited at both sites, which are situated in the accumulation zone. Surface melt at these sites does not result in standing surface water, but rather saturates or percolates below the surface, limiting its effect on surface albedo. Observed early melt season (May/June) surface conditions were a fairly soft and flat snow surface with no sastrugi, drifting, or other surface features.

*Section 3.1 (and elsewhere): When comparing different seasons, I think that the results of the Wilcoxon rank sum tests can be referenced in the tables and don't need to be repeated in each paragraph. The many parentheticals make it a bit challenging to read and the statistics are clearly presented in the tables. Maybe the journal or author preferences dictate that it should also be in each paragraph, but I think it interferes a bit with clarity.*
**[Author response]**
We agree that the parentheticals interfere with clarity and will remove them.

*Figure 4: 1:1 plots of in-situ air temperature vs. MODIS LST are very instructive, and should potentially be included in addition to these box plots.*
**[Author response]**
We will include 1:1 plots of in-situ air temperature vs. MODIS LST

*Lines 247-248: Please indicate if it was tested that the data are normally distributed prior to linear regression, and normalize data if necessary.*
**[Author response]**

Edit line 174: We compare differences between AWS and MODIS LST data to wind speed and solar radiation data obtained from the Divide AWS. We transform LST offsets, wind speed, and solar radiation data to approximately normal distributions using a box-cox transformation and normalize each dataset around zero. We then perform linear regressions on LST offsets vs. wind speed, LST offsets vs. solar radiation, and LST offsets vs. wind speed under low (<400 Wm$^{-2}$) levels of solar radiation.

Regressions on normalized data:

[Figure]

***Figure 8: Adding a horizontal line at 0 C would be helpful in figures a and b; Is this data for all 20 years when Divide AWS data are available?***

**[Author response]**
This is data through 2019, as we did not have ERA5 data for 2020. We will add a horizontal line at 0°C.

***Figure 9: Adding a horizontal line at 0 C would be helpful in figures a and b***
[Author response]
We will add a horizontal line at 0°C

***Lines 287-290: Is it possible that the lack of difference between offsets in MODIS brightness and surface temperatures in the fall and winter is also because of increased inversions relative to other seasons?***
[Author response]
Yes! We chose to focus in this section (4.2) on the implications for emissivity and keep the discussion of inversions contained in section 4.3 for clarity. We will add an explicit statement commenting on the lack of difference in MODIS brightness and surface temperature offsets in fall and winter in relation to inversions either here or after our discussion of inversions below.

***Lines 294-295: I would potentially expect a wider variability in surface conditions (and thus emissivity) in the summer, but this may be site specific. Perhaps explain more why winter could result in more rapidly changing emissivity.***
[Author response]
Updated text to read: Emissivity increases with surface melt, and decreases with increasing particle size and density, which can occur due to either packing or sintering of grains as the snow surface evolves following a snowfall event (Salisbury et al., 1994). At Divide, summertime emissivity changes are likely dominated by alteration of the surface snow by melt, while wintertime emissivity changes are likely dominated by snow surface evolution following snowfall events, which occur more frequently in the winter. The relative magnitude of summer and winter emissivity changes are unknown and may result in the seasonal difference in outcome of the LST algorithm.

***Lines 304-305: The analyses exploring links between accumulation and the LST offset seem to have covered a wide array of options. Was "days since last accumulation" one of the ways this data was analyzed? I couldn't quite tell from the descriptions and that might be a reasonable metric to consider.***
[Author response]
Yes, it was. We've rephrased the statement below to mention this clearly.
Updated text to read: However, given the low temporal resolution of the MODIS data relative to the accumulation record (1 image per day vs. 1 sample per hour), we found no relationship either between the LST offset and individual snowfall events or between the LST offset and the total accumulation each month, the percent of days with accumulation each month, or the mean days between accumulation each month. We also found no relationship between the LST offset and days since last accumulation.

***Lines 314-315: Why used different emissivity values for the different sites, or is this a typo?***
[Author response]
This is a typo. The different emissivities used are end-member snow emissivities both at Divide, not at Divide and Eclipse.

***Line 321: Is it a discrepancy because it is not accounted for given the assumptions used in the T_surface calculations presented here?***
[Author response]
Updated text to read: The smaller magnitude of surface-air temperature offsets at Summit, Greenland and the South Pole relative to our study sites may be due to a stronger influence of turbulent fluxes at Summit and the South Pole or to variations in albedo, as both turbulent fluxes and surface albedo can be strong controls on surface energy balance (Braithwaite and Olesen, 1990; Oerlemans, 1991; Ebrahimi and

Marshall, 2016).

*Lines 323-338: Consider also comparing to 2m near surface inversions investigated in Nielsen-Englyst et al. (2019).*
[Author response]
Updated text to read: In comparing the magnitude of the summer LST offset here (JJA Mdn = 0.98°C), to prior studies, the offsets presented here are smaller than previously observed summer MODIS LST offsets in the St. Elias (5–7°C, Williamson et al. 2017). However, these prior LSTs were daily averages of maximum and minimum values, with most of the offset being attributed to the inclusion of minimum LSTs (Williamson et al., 2017). In contrast, this study uses a single daily LST value and coincident AWS measurements acquired between 11:00 a.m. and 1:30 p.m, when surface and air temperatures are near their maximum, thereby eliminating the effects of any diurnal cycle on observed LST offsets. Our modeled temperature inversions show a diurnal cycle, which is more dramatic in the summer than the winter because of the greater difference between incoming solar radiation during the day and night, and is likely responsible for the higher magnitude of the previously observed summer LST offsets (Fig. 9; Tables 6, 7). A comparison of surface and air temperatures measured in situ over 29 Arctic sites also shows a diurnal cycle that is most pronounced in spring and absent during the winter and polar night (Nielsen-Englyst et al., 2019). The magnitude of the summer LST offset at Eclipse and Divide is in close agreement with temperature inversions observed at Summit, Greenland, where 2 m air and surface temperatures have been contemporaneously measured in situ. During June–July 2015, Summit surface temperatures were 0.32 to 2.4°C lower than 2 m air temperatures (Adolph et al., 2018). At three northern Alaska sites, summer clear-sky surface temperatures were higher than corresponding 2 m air temperatures (Barrow and Atqasuk in 2010, and Olitok Point in 2014; (Good, 2016)). In contrast to sites in Greenland and the St. Elias, these northern Alaskan sites are characterized by seasonal snow cover. Sites with seasonal snow cover present challenges for interpretation because they experience surface melt and a drastic change in surface type over the course of the melt season. Across glaciated areas, sites in the accumulation zone have been found to have the weakest near-surface inversions during the summer, while sites in the ablation zone have been found to have the strongest near-surface inversions during the summer, likely because of the change in surface type with over the melt season (Nielsen-Englyst et al., 2019).

*Line 342-343: Perhaps I'm missing something, but why wouldn't winter surface temperatures also be high then?*
[Author response]
Updated text to read: The dip in modeled summer temperature inversions (Fig. 8) is the result of our 0°C surface temperature cap, which is a simplistic numerical correction for unrealistically high summer surface temperatures above the melting point. Because the 0°C cap is applied after the calculation of surface temperatures and does not address the mechanisms of inversion development, the distinction between capped temperatures slightly over 0°C and uncapped temperatures slightly below 0°C is somewhat arbitrary. We therefore focus on the magnitudes and seasonal patterns of calculated inversions during summer and winter rather than during the shoulder seasons where the temperature cap likely biases our results.

*Lines 370-375: Please explain in a bit more detail how the data are processed before going into the linear regression. I suggest framing the linear fit with the actual variables desired (instead of generic x and y). The corrected MODIS temperatures have a low mean error because of how they are calculated, but how does the standard deviation compare in magnitude to the interannual variability? Is a better fit to the data achieved if it is not averaged annually but instead each paired data point (of 2 m air temp and MODIS*

*LST) is part of the regression?*
[Author response]
We chose not to fit the data using each paired point because our specific goal here was to see if MODIS LSTs could be used to help interpret paleo records on annual timescales.

To be inserted in methods:

2.6 MODIS LSTs and melt

To evaluate whether MODIS LSTs can be used in conjunction with in situ air temperatures to examine the conditions associated with surface melt, we compare interannual trends between the two and reconcile the difference between MODIS LSTs and AWS temperatures using a simple linear regression. We are interested in interannual trends since interpretation of paleo records often occurs on interannual timescales. We therefore calculate the mean annual value for both MODIS LSTs and AWS temperatures. Because we calculate annual means rather than examine individual MODIS LST and AWS temperature pairs, we use all available MODIS LSTs and AWS temperatures, rather than only the subset of dates for which we have both. We fit a linear model to mean annual MODIS LSTs and AWS temperatures using the MATLAB function fitlm(), taking the AWS temperature to be the response variable. We then use the coefficients from this linear fit model to generate a set of converted mean annual MODIS LSTs ($LST_{converted} = -3.35 + 0.49LST$).
The RMSE of our linear fit is 1.88°C, and the interannual variability spans a range of 5.33°C.

***Minor Comments:***

***Line 42: Consider starting a new paragraph with the sentence that begins "Instrumentation…"***
[Author response]
Alternatively, it may make sense to start a new paragraph with the sentence that begins "MODIS LSTs are a valuable tool…". Either way, we agree that breaking this into separate paragraphs improves organization.

***Line 68: Consider phrasing as "apparent MODIS LST offset" here and elsewhere?***
[Author response]
Alternatively, it may improve clarity to use the term "MODIS LST offset" throughout with an explicit statement that surface and air temperatures are different quantities. "Apparent MODIS LST offset" implies that the offset isn't real. If interpreted to mean that MODIS LSTs are not in fact biased, this is true, because surface and air temperatures are different quantities. However, if interpreted to mean that MODIS LSTs and air temperatures should be the same, this is false. Regardless of which phrasing is used, consistency throughout is important for clarity, and that we will revise.

***Line 86: Consider adding a definition of a nunatak***
[Author response]
Updated text to read: (note the many ridges and nunataks, or exposed areas of rock, shown in Fig. 2),

***Lines 88-89: Consider reversing sentence that begins with "Here…" so that it discusses inversions first, which were just being discussed. I found the sentence a bit confusing on the first read because I was unsure which hypotheses were being referenced.***

[Author response]
Updated text to read: Here, we evaluate the plausibility of near-surface temperature inversions in the St. Elias and test alternative hypotheses to explain the offset in MODIS LSTs in the region.

*Line 221 (and elsewhere): forgotten degrees C units on reported median.*
[Author response]
Updated text to read: Mdn = -2:90°C (also edited elsewhere)

*Line 222: "MODIS temperature difference data spaces >10 degrees C" – is this saying the maximum offset is greater than 10 degrees C? Please clarify.*
[Author response]
No, this is referring to the variability of MODIS offsets in each season.

Updated text to read: In all seasons, observed MODIS offsets vary by more than 10°C, with the range of winter values being greatest at 35.56°C at Divide and 25.13°C at Eclipse.

*Line 249: Perhaps an "overwhelming majority" or some other language would be better suited to describe 95%.*
[Author response]
Updated text to read: An overwhelming majority (95%) of MODIS LST offsets…

*Line 256: "modeled temperature difference data is greater than 60 degrees C" – does this mean that the modeled inversions are sometimes up to 60 degrees? Please clarify.*
[Author response]
Updated text to read: In the winter, modeled inversion strength varies by up to 60°C.

*Lines 275-280: Since the manuscript already revealed that the brightness temp and surface temp do show the same pattern, going through the hypothetical scenarios in the way they are framed confused me a bit. Consider rephrasing to make it clear that one of them we already know to be what was observed.*
[Author response]
Updated text to read: We find similar seasonal distributions of offset from AWS temperatures in MODIS LSTs and MODIS brightness temperatures, suggesting that the preferential fall and winter offset is not introduced by the conversion from brightness temperature to surface temperature or the emissivity values used in this conversion (Fig. 5). Moreover, Landsat brightness temperatures also show a pattern of greater offset from AWS temperatures in the fall and winter. The observed apparent cold bias in MODIS LSTs is therefore not unique to the MYD21 product or even the MODIS sensor. Unfortunately, due to the limited availability of ASTER data, too few images exist to examine any seasonal pattern.

*Citations:*

*Guillevic, P., Göttsche, F., Nickeson, J., Hulley, G., Ghent, D., Yu, Y., Trigo, I., Hook, S., Sobrino, J.A., Remedios, J.,Román, M. & Camacho, F. (2017). Land Surface Temperature Product Validation Best Practice Protocol. Version 1.0. In P. Guillevic, F. Göttsche, J. Nickeson & M. Román (Eds.), Best Practice for Satellite-Derived Land Product Validation (p. 60): Land Product Validation Subgroup (WGCV/CEOS), doi:10.5067/doc/ceoswgcv/lpv/lst.001*

*Nielsen-Englyst, P., Høyer, J. L., Madsen, K. S., Tonboe, R., Dybkjær, G., & Alerskans, E. (2019). In situ observed relationships between snow and ice surface skin temperatures and 2 m air*

*temperatures in the Arctic. The Cryosphere, 13(3), 1005-1024.*

**References:**

Adolph, Alden C., Mary R. Albert, and Dorothy K. Hall. 2018. "Near-Surface Temperature Inversion during Summer at Summit, Greenland, and Its Relation to MODIS-Derived Surface Temperatures." *The Cryosphere* 12: 907–20. https://doi.org/10.5194/tc-12-907-2018.

Braithwaite, Roger J., and Ole ? Olesen. 1990. "A Simple Energy-Balance Model to Calculate Ice Ablation at the Margin of the Greenland Ice Sheet." *Journal of Glaciology* 36 (123): 222–28. https://doi.org/10.3189/S0022143000009473.

Ebrahimi, Samaneh, and Shawn J. Marshall. 2016. "Surface Energy Balance Sensitivity to Meteorological Variability on Haig Glacier, Canadian Rocky Mountains." *Cryosphere* 10 (6): 2799–2819. https://doi.org/10.5194/TC-10-2799-2016.

Good, Elizabeth Jane. 2016. "An in Situ-Based Analysis of the Relationship between Land Surface Skin and Screen-Level Air Temperatures." *Journal of Geophysical Research: Atmospheres* 121 (15): 8801–19. https://doi.org/10.1002/2016JD025318.

Herrero, J., and M. J. Polo. 2012. "Hydrology and Earth System Sciences Parameterization of Atmospheric Longwave Emissivity in a Mountainous Site for All Sky Conditions." *Hydrol. Earth Syst. Sci* 16: 3139–47. https://doi.org/10.5194/hess-16-3139-2012.

Hersbach, Hans, Bill Bell, Paul Berrisford, Shoji Hirahara, András Horányi, Joaquín Muñoz-Sabater, Julien Nicolas, et al. 2020. "The ERA5 Global Reanalysis." *Quarterly Journal of the Royal Meteorological Society* 146 (730): 1999–2049. https://doi.org/10.1002/QJ.3803.

Nielsen-Englyst, Pia, Jacob L. Høyer, Kristine S. Madsen, Rasmus Tonboe, Gorm Dybkjær, and Emy Alerskans. 2019. "In Situ Observed Relationships between Snow and Ice Surface Skin Temperatures and 2 m Air Temperatures in the Arctic." *Cryosphere* 13 (3): 1005–24. https://doi.org/10.5194/TC-13-1005-2019.

Oerlemans, J. 2016. "The Mass Balance of the Greenland Ice Sheet: Sensitivity to Climate Change as Revealed by Energy-Balance Modelling:" *Http://Dx.Doi.Org.Wv-o-Ursus-Proxy02.Ursus.Maine.Edu/10.1177/095968369100100106* 1 (1): 40–49. https://doi.org/10.1177/095968369100100106.

Salisbury, John W., Dana M. D'Aria, and Andrew Wald. 1994. "Measurements of Thermal Infrared Spectral Reflectance of Frost, Snow, and Ice." *Journal of Geophysical Research: Solid Earth* 99 (B12): 24235–40. https://doi.org/10.1029/94JB00579.

Staley, D. O., and G. M. Jurica. 1972. "Effective Atmospheric Emissivity under Clear Skies." *Journal of Applied Meteorology and Climatology* 11 (2): 349–56.

Williamson, Scott N., David S. Hik, John A. Gamon, Alexander H. Jarosch, Faron S. Anslow, Garry K. C. Clarke, and T. Scott Rupp. 2017. "Spring and Summer Monthly MODIS LST Is Inherently Biased Compared to Air Temperature in Snow Covered Sub-Arctic Mountains." *Remote Sensing of Environment* 189 (February): 14–24. https://doi.org/10.1016/J.RSE.2016.11.009.

---

## Author Comment (AC2)

Thank you for your constructive comments. We have addressed them line by line below. Comments are indicated by boldface and italicized text; our responses in normal text and preceded by [Author response].

*Overview*

**The manuscript describes a thorough comparison between in-situ 2m air temperature data and a MODIS LST product for two sites in the St Elias mountains. The authors use additional data from ASTER, Landsat, ERA-5 and AWSs to further support their conclusions.**

**The main aim of the study is to determine the cause of the difference between 2-m air temperature measurements and the MODIS LST. Three possible causes are explored: the large MODIS footprint, errors in the surface emissivity, and near-surface temperature inversion. The study finds that the latter is the most likely cause of the temperature offset. The manuscript gives insight into how well MODIS LST represents the actual surface conditions in the remote St Elias mountains, which is important for future monitoring.**

**The manuscript is generally well-written and the presented results are interesting. I have two areas of concern, but most of my comments are minor:**

**1.       I think you should be a bit clearer about the fact that you can't directly compare the 2m air temperature and the land surface temperature. In some sections, you talk about "correcting" the MODIS LST (e.g. Figure 10) – it is not necessarily that the MODIS data is biased, it is just because you are comparing two different things. For the same reason, I would also be careful calling it a "bias" in the title.**

[Author response]
We agree that this language is confusing and will remove the usage of "bias" and "correct" in reference to the MODIS offset and conversion to air temperatures. We will also change our title to "Offset of MODIS land surface temperatures from in situ measurements in the Upper Kaskawulsh Glacier region (St. Elias mountains) indicates near-surface temperature inversions" and insert the following statement preceding our goal to clarify the distinction between air and surface temperatures.

Edited Line 99: Lastly, it may be that the LST offset does not arise during the calculation of LSTs at all, but is a real physical temperature difference between the surface and air due to the development of a near-surface temperature inversion. Although MODIS LSTs can be a useful complement to in situ air temperatures, the two cannot be directly compared and physical differences between the two must be accounted for when using them together.

**2.       You mention that both AWS are situated on nunataks, but you don't really go into detail on the effect of this. If they are placed on nunataks, and not on glacier ice, could this not also be causing part of the bias? See also my specific comments for L 136-138 and L 270.**

[Author response]
See response to comment on L136-138.

**Specific comments**

**L 4-5: is this referring to previous work over St Elias? Or to the current study?**

**[Author response]**

This statement is in reference to prior studies.

Updated text to read: However, MODIS LSTs in the St. Elias Mountains have been found in prior studies to show an offset from available weather station measurements, the source of which is unknown.

*L 39: ""brightness temperature", an intermediate temperature product used to produce the final surface temperature." - I would explain what this is here, not just call it an intermediate product*
**[Author response]**

Updated text to read: Remote sensing temperatures include the final surface temperature product, as well as "brightness temperature", or the temperature of a perfect blackbody emitter under the same conditions.

*Table 1: can you add a bit more info about the different data sources here? Resolution (temporal and spatial) and maybe uncertainty.*
**[Author response]**

We will update Table 1 to contain the below information:

| | Name | Measurement technique | Footprint | Product/instrument | Uncertainty |
|---|---|---|---|---|---|
| Air temperatures | Divide AWS | In situ | Point measurement | Campbell 107F | ± 0.2°C |
| | | | | HOBO S-THB-M008 12-bit sensor | ± 0.21°C |
| | Eclipse AWS | In situ | Point measurement | | |
| | iButton | In situ | Point measurement | Maxim Integrated iButton Data Logger DS1922L | ± 0.5°C |
| Surface temperatures | MODIS LST | Remote sensing | 1 km | MYD21 | |
| | ASTER surface temperature | Remote sensing | 90 m | AST_08 | |
| | MODIS BT | Remote sensing | 1 km | MODTBGA_006 | |
| | ASTER BT | Remote sensing | 90 m | Calculated from ASTL1T following Ndossi and Avdan (2016) | |
| | Landsat BT | Remote sensing | Resampled from 100 m to 30 m (Landsat 8) | LC08 | |
| | | | Resampled from 120 m to 30 m (Landsat 5,7) | LE07 LT05 | |

*L 109: What do you mean with "more influential"? In terms of current sea level rise?*
**[Author response]**

Updated text to read: Importantly, Alaska contains much more glacial ice than the European Alps, and therefore has a larger potential contribution to global sea level rise.

*L 111: you do not define "Divide Icefield" as "Divide" until later in the text*
**[Author response]**

Edited L108: The AWS record from Icefield Divide (hereafter referred to as "Divide") is, to our knowledge, the longest such record from a glaciated high alpine area outside the European Alps.

*L 125-126: How do you know the datasets are consistent, when the time periods do not overlap? Please clarify.*
[Author response]
We did not perform a robust test of the datasets' consistency, instead choosing to focus on data from Divide, which is more abundant. However, we still included what data we had from Eclipse to supplement the results at Divide.

Updated text to read: We combine the Eclipse AWS and iButton datasets for maximum coverage at the site.

*Figure 2: Where is the location of the iButton?*
[Author response]
The iButton was located at the same site as the AWS. We will add this to the legend and caption.

*Table 2: change "Ice core site – AWS site" to "MODIS Ice Core Site – MODIS AWS site", to clarify it is not in situ observations.*
[Author response]
Edited table heading: "MODIS Ice Core Site – MODIS AWS site"

*L136-138: I am not sure I follow this. You are comparing two MODIS pixels – one with only ice, and one with ice and a nunatak, to find out the difference in temperature between the nunatak surface and the ice surface? If so, this is interesting, but should be clarified and mentioned in the discussion. In addition, I would guess that the difference between the AWS and the ice covered ground is bigger than found in this comparison, since both MODIS pixels does contain some glaciated area.*
[Author response]
Yes, here we are comparing two MODIS pixels – one with only ice, and one with ice and a nunatak. However, our purpose is not really to find out the difference in temperature between the nunatak surface and the ice surface, but rather simply to see if there is a discernible difference. We are interested in the ice surface, so if there is a difference in the MODIS pixel with only ice vs. ice and rock, we want to mitigate the effect of the rock in our subsequent analysis.

Updated text to read: Our goal is to determine the dominant source of the offset in MODIS LSTs at glaciated sites in the St. Elias. Because the Eclipse and Divide AWS are located on nunataks, we test for the LST offset using MODIS data encompassing adjacent ice core sites ~3 km from each AWS location, thereby excluding the dark nunatak surface from the MODIS pixel and focusing on the ice surface (Fig. 2). We compute the difference in MODIS LST between the ice core site grid cell (containing only ice) and the AWS site grid cell (containing ice and rock) to determine whether the inclusion of the nunatak has a discernible effect on the MODIS LST.

Insert in results section (and add figure plotting temperature differences between AWS and ice core sites): MODIS data at the Divide AWS nunatak and adjacent ice core site have a median temperature difference of 0.86°C and interquartile range of 1.97°C. The difference between the two sites shows greater variability in the fall (IQR = 3.21) and winter (IQR = 3.98) than in the spring (IQR = 0.70) and summer (IQR = 1.80), with the ice core site tending to be slightly colder in the winter (median temperature difference of -0.49°C), but warmer in the spring (median = 0.95°C), summer (median = 1.33°C) and fall (median = 0.28°C). Temperature differences between the Divide AWS and ice core site are summarized in Table 2.

Note: Table 2 will be edited to show differences between the ice core and AWS sites as medians rather than means so as to be more directly comparable with median LST offsets from AWS temperatures.

To be inserted in the discussion section: MODIS LSTs at the ice core site do not tend to be colder that at the AWS site except during the winter. The inclusion of the warmer nunatak surface in the MODIS grid cell at the AWS site fails to provide a compelling explanation for the colder wintertime LSTs at the ice core site, given that more of the rock surface would likely have snow cover during the winter. The colder wintertime LSTs at the ice core site may contribute to the MODIS LST offset from in situ temperature measurements examined in this study. However, this contribution is too small (median = -0.49°C) to explain the magnitude of the MODIS LST offset at the Divide ice core site (median = -8.40°C). In the spring, summer, and fall, the LSTs at the ice core site tend to be slightly warmer than at the AWS site. Results here may therefore underestimate the magnitude of the MODIS LST offset from AWS temperatures in these seasons.

*L 144: why only between 11 and 1:30?*
[Author response]
MODIS overpass times are all from approximately 11:00-13:30. We choose to further narrow our data acquisition to the hours for which the viewing angle of our site is < 30°, which are from 12:00-13:00. Updated text to read: Temperature differences between the Divide AWS and ice core site are summarized in Table 2. MODIS LST data were obtained for the period 2000-2020 (https://lpdaacsvc.cr.usgs.gov/appeears/) for dates with minimal cloud cover between the hours of 12:00 and 13:00, when viewing angle is less than 30°, to mitigate the effect of viewing angle on temperature and emissivity. At Divide, 742 MODIS images spanning 2002-2020 were analyzed. Seasonally, 203 images were acquired in spring (MAM), 169 in summer (JJA), 188 in fall (SON), and 182 in winter (DJF). The average time between scenes at Divide was 9 days after filtering. At Eclipse, 100 MODIS images were analyzed: 87 spanning June 2005 through June 2007 and 13 spanning November 2016 through February 1017. Each MODIS image was paired with the closest hourly measurement available in the AWS data.

*L 145-147: Do I understand correctly, that the 700+ images at Divide span 20 years, and the 100 images at Eclipse span ~2 years of data?*
[Author response]
See edited statement for L144 above

*L 199: How do you get the downward radiation for Eclipse/iButton?*
[Author response]
We only apply the simple energy balance model at Divide because of the lack of data at Eclipse. L339-0340 contains a typo including Eclipse and is edited below:

Results from the simple energy balance model predict no summertime inversion at all, with surface temperatures being a median of 0.77°C higher than 2m air temperatures using an emissivity value of $\varepsilon_S = 0.95$ and 0.75°C higher than 2m air temperatures using an emissivity value of $\varepsilon_S = 0.99$.

*L 200: can you provide a bit more info about the ERA-5 product you use?*
[Author response]
Updated text to read: We calculate downward longwave radiation as follows, using 2 m air temperature ($T_a$) from Divide and atmospheric emissivity ($\varepsilon_a$) derived from the ERA5 reanalysis longwave radiation product. We use only the derived emissivity from the ERA5 product, rather than the total downward radiation in order to use measured values (in situ 2 m air temperature) where possible. ERA5 outputs have

a spatial resolution of 31 km; data span 2002-2019 every six hours (Hersbach et al., 2020). Atmospheric emissivity increases with increasing surface vapor pressure (Staley and Jurica, 1971). Our atmospheric emissivity values ranged from ~0.48 to 1. Atmospheric emissivity measured over the Sierra Nevada (Spain) from 2005-2011 ranged from ~0.4-1 (Herrero and Polo, 2012).

*Page 10, 11 and others: consider your number of significant digits – are your results really that accurate? I would stick to 1-2 significant digits. Also in e.g. L 255: if it is a simple model, it probably does not have an accuracy of 3 significant digits.*
[Author response]
Significant digits will be reduced to 2 throughout

*Figure 6: this is at Divide?*
[Author response]
Yes, this is at Divide

Edited figure title: Comparison of the MODIS LST offset (MODIS-AWS) with measured solar radiation and wind speed at Divide.

*L 270: Why do you compare MODIS, Landsat and Aster over the ice core location and not the AWS location (if I understand figure 2 correctly). If you are investigating the cause of the difference in AWS and measured LST, it would make more sense to look at the AWS location – especially since the AWS is on a nunatak, you would be able to better investigate the effect of this.*
[Author response]
We compare MODIS, Landsat and ASTER over the ice core location because our motivation is to use remote sensing methods to obtain temperature records over large ice-covered areas and we therefore do not want to focus on just the nunatak. See response to comment on L136-138. We initially tested the footprint and emissivity hypotheses using all three sensors. Although we used AWS temperatures to calculate the offset of each remote sensing product, these tests essentially involved comparing remote sensing products against one another. When we began to address the possibility of a near-surface temperature inversion, we then incorporated MODIS LST data from the AWS site because it is a more direct comparison between surface and air temperatures at the same site. We did not feel it necessary to re-test the footprint and emissivity hypotheses with remote sensing data over the AWS site because: a) the temperatures between the sites are very similar and produce the same seasonal distribution of MODIS LST offsets, and b) whether we use the ice core site or the AWS site to calculate offsets for each sensor, it is the relationship between sensors that is pertinent for the footprint and emissivity hypotheses, and as we had already eliminated those hypotheses, we do not investigate them in further detail as we do near-surface temperature inversions.

*L 302: How is the DIVIDE snowfall record measured? Maybe give some information about this in the data section.*
[Author response]
To be inserted in section 2.1 Study sites and in situ data:

The Divide snow accumulation record was obtained using a Campbell Scientific SR50 ultrasonic snow depth sounder instrument. The instrument provided twice-daily readings of its distance from the snow surface at the Icefield Discovery Camp during the period spanning 2003-2012, corrected for the variability in speed of sound with air temperature.

*L 314-315: Why are you using different emissivities for the two sites?*
[Author response]

This is a typo. The different emissivities used are end-member snow emissivities both at Divide, not at Divide and Eclipse.

***Figure 10: What happened in 2020? The AWS temperature is much lower than the MODIS temperature.***
[Author response]
Our data for 2020 were incomplete at the time of acquisition, only running through June, so the warmest months were not included. For clarity, 2020 will be omitted.

**References:**

Herrero, J., and M. J. Polo. 2012. "Hydrology and Earth System Sciences Parameterization of Atmospheric Longwave Emissivity in a Mountainous Site for All Sky Conditions." *Hydrol. Earth Syst. Sci* 16: 3139–47. https://doi.org/10.5194/hess-16-3139-2012.

Hersbach, Hans, Bill Bell, Paul Berrisford, Shoji Hirahara, András Horányi, Joaquín Muñoz-Sabater, Julien Nicolas, et al. 2020. "The ERA5 Global Reanalysis." *Quarterly Journal of the Royal Meteorological Society* 146 (730): 1999–2049. https://doi.org/10.1002/QJ.3803.

Staley, D. O., and G. M. Jurica. 1972. "Effective Atmospheric Emissivity under Clear Skies." *Journal of Applied Meteorology and Climatology* 11 (2): 349–56.

---

## Author Comment (AC3)

Thank you for your very thorough review of our paper. We have addressed your general comments in a bulk response and your specific comments line by line below. Comments are indicated by boldface and italicized text; our responses in normal text and preceded by **[Author response]**.

*This manuscript should be rejected for publication.*

*General comments:*

*The major finding seems to be the difference in air temperature and glacier temperature is the result of an inversion. Surface temperatures of snow and ice are usually colder than air temperatures. This is particularly true during the melt season and is a pretty well established fact. I am unsure what this manuscript offers that is not already present in the literature.*

*The aims and goals should be refined and better described in the manuscript. The organisation of the manuscript requires substantial revision.*

*No lapse rates are reported, which is the typical way to identify an inversion.*

*Editing for clarity is required. Details need to be added to the many vague statements in the manuscript.*
*Typically these sorts of studies use orders of magnitude more data than what appears here. An argument needs to be presented that the small data set is adequate. Static (or literature) values need to be replaced with measurements where possible.*

*The placement of figures in the narrative is disjointed and not logical. Many results are being presented in the discussion.*

*Many references concerning the relationship between energy balance and glacier mass balance are missing.*

*There is substantially more meteorological data available than what has been used in this analysis.*

**[Author response]**
    The contribution of our work is providing a comparison of air and surface temperatures in an understudied region containing a large amount of glaciated area. Although it is known that surface temperatures of snow and ice are often colder than air temperatures, there are few studies that examine this phenomenon in alpine terrain or address its relevance to monitoring alpine glaciers and interpreting alpine paleoclimate records. Additionally, our data suggest that near-surface inversions are more pronounced in the winter, not during the melt season, and thus warrant further examination. As the importance of regional variability becomes ever clearer from a glacier mass balance perspective, the limitations of current temperature monitoring in the St. Elias mountains require attention. Our focus here is on MODIS LSTs as a tool to understand a vulnerable and rapidly changing region, rather than on the St. Elias mountains as a case study to examine MODIS LSTs and air temperatures. We have framed our goal to address three specific hypotheses (footprint, emissivity, inversion) about the source of the offset between MODIS LSTs and in situ air temperatures in the St. Elias mountains. Each hypothesis is addressed in its own set of methods/results/discussion subsections. We have reorganized portions of the manuscript for clarity. Specifically, we have ensured that each subsection in the results and discussion is preceded by a corresponding subsection in methods, and rearranged the figures accordingly. We will add

statements making the link between our three hypotheses and monitoring temperature/interpreting paleoclimate records more explicit, as this appears to be the most unclear aspect of our goal and motivations based on comments above.

Regarding clarity and vague statements, please see our responses to comments below, specifically, L21, L29, L96, L130, L367. Please also see responses to comments below for added references. We've used measurements wherever possible; measurements are extremely limited in this region, which is a large motivation for the study. Regarding lapse rates, although lapse rates are typically used to report large-scale inversions, the same is not true for near-surface inversions, which is what we are specifically dealing with here. We will add a statement clarifying that our use of the term "inversion" specifically refers to near-surface inversions throughout the paper.

We use the longest meteorological record ever compiled for the St. Elias for our analysis, and indeed one of the longest such records when considering alpine regions globally (please see response to comment on Line 108 below). Williamson et al. (2020) summarize the meteorological stations in the St. Elias region. Out of 16 stations, 7 are located on glaciated terrain. The others are on talus, alpine tundra, grass or gravel. Six out of the 7 stations on glaciated terrain are located above 2500 m. None of the non-glaciated station sites are above 2500 m. Four of the 6 stations on glaciated terrain above 2500 m have records less than or equal to two years in length. The remaining two are the stations we use at Divide, giving a combined record of 20 years. Although the Eclipse record is too short to produce a robust dataset on its own, we include data from Eclipse in our analysis because of its proximity to Divide. We do not include the other three (Ogilvie Glacier, King Col, and Prospector-Russel Col) because their records are likewise too short to produce robust datasets.

***Specific comments:***

***Abstract:***

***MODIS LST can also be sparse or absent***
[Author response]
When we say that in situ measurements are sparse or absent in the St. Elias, we mean that there are very few measurements distributed across a large area. Although a dataset of MODIS LSTs may not have very good temporal resolution (since the surface is often obscured by cloud cover), they do provide measurements in space that realistically cannot be obtained using in situ methods because of the challenges associated with installing and maintaining equipment in the remote mountainous environment.

***MODIS LSTs are offset… which each LST measurement, average LST, minimum, maximum…***
[Author response]
This is referring to each individual LST measurement as compared to the closest hourly weather station measurement.

***Footprint usually refers to swath width, or some derivative. Is it the grid cell size you are referring to?***
[Author response]
Yes, we are referring to the grid cell size.

***Snow emissivity is >0.8 and can be close to being a blackbody, so it is intuitive that brightness temperature would also contain bias.***
[Author response]

It may be intuitive, but we consider this important to explicitly address and report since the intuitive answer to a question may not always be the right one.

***Line 21: …with far reaching impacts. This is the kind of statement that the manuscript is peppered with and is virtually meaningless: please revise, here and throughout.***
**[Author response]**
Updated text to read: In recent decades, the high latitudes (>60°) have warmed at a more rapid rate than the rest of the planet, with impacts extending to distant lower latitude regions (Winton, 2006; Serreze and Barry, 2011; You et al., 2021).

We have edited the statement to be more specific in referring to impacts in lower latitude regions, as the point of this statement is that high latitude temperature changes are relevant to processes across other parts of the globe. We believe this is pertinent context for studying temperature changes in the high latitudes.

***Line 23: reduced the Earth's albedo, further accelerating warming. Please provide credible references for this statement. Most studies do exactly what you are doing which is confusing correlation and causation. Perhaps as the temperature increases more snow is melted, and the newly exposed area provides a negligible amount of atmospheric warming. For context read: https://www.nature.com/articles/s41598-018-27348-7. Alternatively, the reduction in snow and ice causes a warming, but the amount of increase in temperature cannot be disentangled from warm air advection. Alternatively, the snow albedo feedback melts glaciers pretty efficiently.***
**[Author response]**
Updated text to read: In particular, the loss of Arctic glaciers has reduced the Earth's albedo (which can further accelerate warming) and contributed to global sea level rise (Budyko, 1969; Lian and Cess, 1977; Serreze and Barry, 2011; Zemp et al., 2019; Hugonnet et al., 2021).

***Line 23-24: As written this statement is not correct. Hugonnet et al.(2021) didn't analyse albedo, nor was it mentioned in Zemp.***
**[Author response]**
See above for added references.

***Line 25: Some ink should probably be spilled on your geographical definition of the Arctic. From a climatology point of view (i.e., Arctic Amplification) Arctic is defined as north of the Arctic Circle.***
**[Author response]**
To avoid misunderstandings with the use of "Arctic", we will change our language to "high-latitude", and define this explicitly as above 60°N.

***Line 29: I don't know how many crucials and criticals I have seen to this point. The writing will pack more punch if these types of words are used substantially less often.***
**[Author response]**
Updated text to read:

Line 28: Additionally, Alaskan glaciers are losing mass at some of the highest rates globally (~66.7 Gt yr$^{-1}$), and therefore remain pertinent to projections of global sea level rise (Hugonnet et al., 2021). The greater North Pacific cordillera contains over 40 mm of global sea level rise potential in a combination of large icefields and small alpine glaciers, making widespread monitoring of glacier mass in the region a worthwhile endeavor (Farinotti et al., 2019).

Line 56: Remote sensing temperature products are especially useful for relating glacier behavior and mass balance to climatological changes in rugged alpine regions where glaciers tend to be at higher elevations than most nearby weather stations.

Line 364: We recommend continued work to understand near-surface thermal processes in these complex regions, including obtaining in situ air and surface temperatures to validate these results.

Line 365: Despite some uncertainty about the exact mechanism for the MODIS offset, and the lack of an accurate physical correction, MODIS LSTs can still shed light on the  question of surface melt and mass balance in the North Pacific…

Line 411: This work provides a  step forward in using remote sensing imagery to expand in situ records and thus provide insight into past and present temperature changes in the St. Elias Mountains and broader North Pacific region.

***Line 32: controlled by atmospheric warming: not necessarily true, these might simply be correlated.***
**[Author response]**
Updated text to read: Glacier mass changes are driven by changes in the surface energy balance; in effect, glacier mass loss is largely associated with atmospheric warming, which relates to surface energy balance through its influence on downward longward radiation and sensible heat transfer (Cuffey and Paterson 2010).

***Line 33: continued -> projected?***
**[Author response]**
Updated text to read: In order to better predict the impacts of continued atmospheric warming, we need to monitor temperature change and glacier response.

***Line 34: delete "to be able"***
**[Author response]**
See above response to comment on Line 33

***Line 38: What does "Remote sensing temperatures include the final surface temperature" mean?***
**[Author response]**
Updated text to read: A variety of temperature products can be obtained using remote sensing methods, including the final surface temperature product, as well as "brightness temperature", or the temperature of a perfect blackbody emitter under the same conditions.

***Line 44: high temporal resolution and long temporal record; they provide two decades… what resolution, which decades? Always provide dates, rates, numbers, values, colours, weights, dimensions, etc. when describing quantitative subjects.***
**[Author response]**
Updated text to read: MODIS LSTs are a valuable tool for monitoring climate in remote regions because they provide more than two decades (2000-present) of near-daily imagery under clear-sky conditions.

***Line 59: "Lower elevation sites receive moisture from different air masses" detail why this is important***
**[Author response]**

It is important that low and high elevation sites are in contact with different air masses because this means data from a low elevation site cannot be used to represent a high elevation site and vice versa. The moisture content is not the key piece here and is removed in the edited statement below.

Updated text to read: Lower elevation sites are in contact with different air masses and are sensitive to different sources of variability than their high elevation counterparts, so data from these stations are not necessarily representative of climatic behavior at glaciated alpine sites (McConnell, 2019).

***Line 65: Not necessarily a universal phenomenon:  see: https://link.springer.com/article/10.1007/s00704-012-0687-x.***

**[Author response]**

Updated text to read: Modeling studies (Chen et al., 2003; Giorgi et al., 1997) predict that warming rates increase with elevation. Although not a universal phenomenon (Ohmura, 2012), elevation-enhanced warming has been observed in a number of alpine mountain ranges including the St. Elias and greater North Pacific cordillera (high elevation sectors of Alaska and parts of the Yukon and British Columbia; Williamson et al., 2020; Diaz et al., 2014; Pepin et al., 2015; Rangwala and Miller, 2012).

***Line 71: "surface itself" should be replaced with details like where the photons are being emitted e.g., from the top x nm of the snow and ice, etc.***

**[Author response]**

We use the term "surface itself" because "surface" is the standard term used in the literature and is sufficient information to contrast with the air. Something like the top x nm of the snow and ice would be more important language if our point were to contrast the surface with deeper snow/ice.

***Line 75:  This paper is relevant here: https://journals.ametsoc.org/view/journals/clim/26/5/jcli-d-12-00250.1.xml. There is probably only a very minor contamination issue.***

**[Author response]**

Updated text to read: Without accurate cloud masking, apparent cold biases in MODIS LSTs have been previously observed at Summit, Greenland in both summer (3°C; Koenig and Hall, 2010) and winter (5°C; Shuman et al., 2014). However, the cloud mask has since been updated to address this problem (Yao et al., 2020). Additionally, previous work in the St. Elias mountains indicates that MODIS LSTs over warm (>0°C) surfaces are an average of <2°C (Williamson et al., 2013).

***Line 77: Summit should have Greenland appended to it, here and elsewhere, when referring to the summit of GIS.***

**[Author response]**

We will add "Greenland" to all usages of "Summit".

***Line 80:  More detail is required here:  There is more forcing that downwelling solar.  Air parcel advection plays a role.  And why does it have to be balanced- the temperature might be changing? Provide rationale.***

**[Author response]**

Updated text to read: Near-surface temperature inversions occur when the surface is colder than the air directly above it and develop over glaciated regions when heat transfer from the surface to the air occurs as a result of an energy imbalance at the surface-air interface (Adolph et al., 2018). Such energy imbalances can occur under low incoming solar radiation, when upward longwave radiation emitted by the earth's surface may exceed downwelling energy fluxes (Adolph et al., 2018). Snow surfaces often have a high emissivity (0.949-0.997 in the 10.5-12.5 μm range; Hori et al. 2006) relative to the

atmosphere, which has been observed to be as low as ~0.4, depending on water vapor content (Herrero and Polo 2012). This difference in emissivities requires the snow surface to cool relative to the air above as it equilibrates (Hudson and Brandt, 2005).

*Line 82: efficient emitter than the atmosphere - implies the atm has a lower emissivity than snow surface. Provide details. Atmospheric emissivity is mainly dependent on water vapour concentration.*
[Author response]
See edited statement above for line 80

*Line 92: pixel is a picture element of a computer screen, where the minimum resolution is set by the screen parameters. Using pixel to describe a remote sensing array element or grid cell is common usage, but not technically correct.*
[Author response]
We will replace all uses of "pixel" with "grid cell".

Updated text to read: The heterogeneity of the St. Elias' environment (surface type, elevation, aspect, incline, wind scouring, shading) may not be well represented by the average temperature value of a MODIS grid cell.

*Line 96: How exactly would "disparate changes in emissivity" lead to a bias? Provide details.*
[Author response]
Updated text to read: Therefore, the icefields undergo disparate changes in emissivity over hours to days, meaning that identifying a single representative emissivity value is challenging. Employing too high an emissivity value in the calculation of LST would result in too low a surface temperature.

*Line 108: There are records longer in the Tibetan Plateau and on Greenland and very possibly elsewhere. For some context see: Global Historical Climatology Network Monthly—Version 3 (GHCN-Mv3) (www.ncdc.noaa.gov/data-access/land-based-station-data/land-based-datasets/global-historical-climatology-network-monthly-version-3). The GHCN-Mv3 ftp server provides a list of weather stations (ftp://ftp.ncdc.noaa.gov/pub/data/ghcn/v3/products/ghcnm.v3.first.last) with associated country codes, station location, elevation, and data duration.*
[Author response]
Global

[Figure]

Plotted above are all of the GHCN weather stations located higher than 2500 m a.s.l. with records longer than 20 years (the length of the Divide record). The Antarctic stations are not relevant here, as we focus on alpine areas and not ice sheet plateaus. Many stations in the western U.S. (and likely elsewhere) are likewise irrelevant, as they are not located in glaciated terrain (see station sites plotted atop RGI 6.0 glaciers by region below). Even without removing the Antarctic and non-glaciated locations, the AWS record at Divide is the only high-elevation weather station record of its length in Alaska, as well as northern Canada and the Russian Arctic.

Western Canada/US

[Figure]

Asia

[Figure]

Europe

[Figure]

Central America

[Figure]

South America

[Figure]

Caucasus/ME

[Figure]

Updated text to read: The AWS record from Divide is, to our knowledge, the longest such record from a glaciated high alpine area in Alaska and the surrounding area.

***Figure 3: Landsat has different sensors (MSS, TM, ETM+, etc.) so either break these up in the figure or identify differences in the text/caption, or both.***
[Author response]
We used LS8 OLI-TIRS, LS7 ETM+, and LS5 TM. We will break these up in the figure/text/caption.

***Line 119:  Is air temp. samples on the hour of hourly averages of sub-hour measurements?  MODIS LSTs are essentially samples.***
[Author response]

Updated text to read: In situ temperatures at Divide were obtained from two adjacent AWS located on small nunataks, the first of which used a Campbell 107F temperature probe (±0.2°C) housed inside a solar radiation shield, which recorded hourly readings from 2002-2015. The second AWS was located ~300 m from the first, and recorded hourly temperatures with a HOBO S-THB-M008 12-bit sensor (±0.21°C) housed inside a solar radiation shield from 2009-present (Fig. 3). Both sensors at Divide were located approximately 2m above the surface. The height of sensors above the surface changed with snow accumulation; however, accumulation on nunataks at Divide is typically limited by intense wind scouring so the sensor height above the surface remains relatively constant over time. Both sensors collected temperature data as hourly averages of 5 minute sampling intervals (Williamson et al., 2020).

***Line 122: Not correct. As snow level changes the Divide sensor's height above the surface will change. It is possible that it also gets buried in some of the winter months.***
**[Author response]**
See response to comment on Line 119 above.

***Line 124: "plastic container"? Provide details. Was this vented passively? Exposed to direct sunlight?***
**[Author response]**
Updated text to read: Available temperature data at Eclipse are lower quality than at Divide, with limited temporal coverage and sensors not up to World Meteorological Organization standards. We therefore focus on Divide, but include available data from Eclipse with the caveat that results are less robust. Temperatures at Eclipse were obtained from an AWS from 2005-2007, and a Maxim Integrated iButton Data Logger DS1922L (  0.5°C) from 21 May 2016 to 17 May 2017, both located on or near a bedrock outcrop ~3 km from the site of an ice core drilled at Eclipse in 2016 (Fig. 3). The AWS recorded hourly averages of subhourly sampling intervals using digital sensors housed in a passively vented radiation shield at a height of approximately 2 m (Williamson et al., 2020). The iButton recorded temperatures at 3-hour intervals and was placed inside an unvented plastic container shielded with rocks. Because data is so limited at Eclipse, we combine the AWS and iButton datasets for maximum coverage at the site. We refer to both the Divide AWS and the combined Eclipse iButton and AWS data as "AWS" for the remainder of this paper.

***Line 125: We combine the Eclipse AWS and iButton datasets… Why? Is this a valid method? Provide sensitivity analysis.***
**[Author response]**
We did not perform a robust test of the datasets' consistency, instead choosing to focus on data from Divide, which is more abundant. Around 88% of the temperature data used in this study came from Divide. Additionally our examination of other meteorological variables and our surface energy balance calculations are all performed with data from Divide. However, we still included what data we had from Eclipse to supplement the results at Divide. See edited statement above in response to comment on Line 124.

***Line 126: consistent. - define, preferably statistically.***
**[Author response]**
See above

***Line 130: "employ an improved method" provide details and why relevant here.***
**[Author response]**
Updated text to read: The MOD21 and MYD21 (together referred to as MxD21) products dynamically retrieve emissivity values for each grid cell, rather than assigning them based on land cover as was done

for the MxD11 products previously examined (Williamson et al., 2017; McConnell, 2019), and have been shown to correct for MxD11 apparent cold biases over barren, but not glaciated, surfaces (Hulley, 2017; Li et al., 2020; Yao et al., 2020).

This change in emissivity assignment is relevant for clarification that the MODIS LSTs we are using are not exactly the same product as those used in prior studies.

***Line 135 (and below): It appears results are provided before methods have been described. It is not clear what is being compared. Is it daily averages of air temperature? Have temperatures (air and MODIS) been temporally matched?***

**[Author response]**
Table 2 has been moved to the results section.

Updated text to read: Our goal is to determine the dominant source of the offset in MODIS LSTs at glaciated sites in the St. Elias. Because the Eclipse and Divide AWS are located on nunataks, we test for the LST offset using MODIS data encompassing adjacent ice core sites ~3 km from each AWS location, thereby excluding the dark nunatak surface from the MODIS pixel and focusing on the ice surface (Fig. 2). We compute the difference in MODIS LST between the ice core site grid cell (containing only ice) and the AWS site grid cell (containing ice and rock) to determine whether the inclusion of the nunatak has a discernible effect on the MODIS LST. MODIS LST data were obtained for the period 2000-2020 (https://lpdaacsvc.cr.usgs.gov/appeears/) for dates with minimal cloud cover and a viewing angles < 30°, to mitigate the effect of viewing angle on temperature and emissivity. At Divide, 742 MODIS images taken between 11:00 a.m. and 1:30 p.m. were analyzed. Seasonally, 203 images were acquired in spring (MAM), 169 in summer (JJA), 188 in fall (SON), and 182 in winter(DJF). The average time between scenes at Divide was ~9 days after filtering. At Eclipse, 100 MODIS images taken between 11:00 a.m. and 1:30 p.m. were analyzed. Seasonally, 25 images were acquired in spring, 24 in summer, 29 in fall, and 22 in winter. The average time between scenes at Eclipse was ~43 days after filtering. MODIS LSTs were subtracted from the nearest hourly in situ air temperature measurement to calculate their offset from in situ temperatures. A small number of summer MODIS LST offset results were skewed by air temperatures well above 0°C (30 dates with air temperature > 4°C, 5 dates with air temperature >8°C), as the snow surface cannot warm above freezing without melting. Removing these dates reduced the temporal coverage of the summer MODIS LST offset data, but had no effect on the seasonal distribution of the offset.

Move to results section: MODIS data at the Divide AWS nunatak and adjacent ice core site has a mean temperature difference of 0.27°C and standard deviation of 2.20°C. The difference between the two sites shows greater variability in the fall (std = 140 2.64) and winter (std = 2.95) than in the spring (std = 1.44) and summer (std = 0.77), with the ice core site tending to be slightly colder (mean winter temperature difference of -0.80°C). This may be due to the inclusion of the warmer nunatak surface in the MODIS pixel at the AWS site. Temperature differences between the Divide AWS and ice core site are summarized in Table 2.

***Line 141: "This may be due to the inclusion of the warmer nunatak surface" - this is testable by comparing time series from grid cells which contain less (or none) exposed rock.***

**[Author response]**
Yes, in lines 135-142 we are comparing two MODIS grid cells – one with only ice, and one with ice and a nunatak. Our purpose is to see if there is a discernible difference between the two. We are interested in the ice surface, so if there is a difference in the MODIS pixel with only ice vs. ice and rock, we want to mitigate the effect of the rock in our subsequent analysis. We find that there is a difference, and the

statement "this may be due to the inclusion of the warmer nunatak surface" is a possible explanation. We reorganize this (see above response to comment on Line 135) to separate our methods and results more clearly.

***Line 144: What is the rationale for using only <30 degrees view angle?  Is there a sensitivity analysis or a citation to confirm this?***
**[Author response]**
The coefficients in the MODIS LST algorithm vary with viewing angle; the RMSE between calculated and modeled surface brightness temperatures (which are used in MODIS LST calculations) increases exponentially with view angle (Hulley et al., 2016). The RMSE is <1°C for viewing angles <60°; we cap our viewing angles at 30° to be conservative.

Updated text to read: Temperature differences between the Divide AWS and ice core site are summarized in Table 2. MODIS LST data were obtained for the period 2000-2020 (https://lpdaacsvc.cr.usgs.gov/appeears/) for dates with minimal cloud cover between the hours of 12:00 and 13:00 (local solar time), when viewing angle is less than 30°, to mitigate the effect of viewing angle on temperature and emissivity (Hulley et al., 2016). At Divide, 742 MODIS images spanning 2002-2020 were analyzed. Seasonally, 203 images were acquired in spring (MAM), 169 in summer (JJA), 188 in fall (SON), and 182 in winter (DJF). The average time between scenes at Divide was 9 days after filtering. At Eclipse, 100 MODIS images were analyzed: 87 spanning June 2005 through June 2007 and 13 spanning November 2016 through February 1017. Each MODIS image was paired with the closest hourly measurement available in the AWS data.

***Line 145: This temporal subset will sample somewhere below the maximum daily temperature.  This also seems to be a very small amount of data from what should be available from a 20 year time series, from two sensors and multiple daily overpasses.***
**[Author response]**
We chose this temporal subset because that is when we could obtain the most MODIS overpasses with low viewing angles that could be matched with close in situ measurements. We will include a comment that this is sampling below the maximum daily temperature. Unfortunately data acquisition is severely limited by cloud cover, which is why a 20 year timeseries yields such limited data.

***Line 146: "The average time between scenes" describe what this means and why it is important - as written I have no idea what it means.***
**[Author response]**
Updated text to read: The mean time elapsed between two consecutive analyzed images at Divide was nine days.

***Line 151: Removing these data ?***
**[Author response]**
Updated text to read: Removing these data reduced the temporal coverage of the summer MODIS LST offset data, but had no effect on the seasonal distribution of the apparent bias.

***Line 160: Under development as of when?***
**[Author response]**
Updated text to read: Landsat surface temperatures remain under development (as of July 2021) and were therefore not included in this study.

***Line 167: TOA Tb is not really a useful metric to compare to surface temperature.***
**[Author response]**
We included the Landsat TOA brightness temperature because it was the closest available product used in the calculation of surface temperatures. However, if its inclusion is more confusing than useful, we can remove it.

***Line 170-175: Provide bounding values for "low" and "high".***
**[Author response]**
Updated text to read: To test whether the MODIS LST offset reflects pervasive near-surface temperature inversions, we examine whether the offset is more pronounced under conditions that facilitate near-surface inversions, namely low levels of incoming solar radiation and low wind speeds. At Summit, Greenland, no inversions greater than 2°C were observed in the 2 m above the snow surface when incoming solar radiation was above 600 W m$^{-2}$ or wind speed was greater than approximately 7 m s$^{-1}$ (Adolph et al., 2018). Over 22 sites in Greenland, maximum temperature inversions were observed at wind speeds of 5 m s$^{-1}$

***Line 176: "would" -> could.***
**[Author response]**
Updated text to read: To test if a near-surface temperature inversion could occur under surface conditions at Divide and Eclipse, we compare the MODIS LST offset to the offset from AWS temperatures of surface temperatures calculated using the following simple energy balance model. The net surface energy balance ($E_N$) can be expressed by…

***Line 176: What does "physically plausible under surface conditions" mean exactly?***
**[Author response]**
See edit for line 176 above

***Line 177: "theoretical model of temperature inversions.To" - Provide details and a space after the period.***
**[Author response]**
See edit for line 176 above

***Line 185: Typically the terms you avoid are small compared to the dominant terms you include. Provide a range of values for all the terms. This will allow the reader to evaluate the effect of removing some terms.***
**[Author response]**
Updated text to read: We ignore $E_G$ because it is often small relative to both radiative and turbulent fluxes, and several studies (e.g. Brock and Arnold, 2000; Hock and Noetzli, 1997; Favier et al., 2004) have validated energy models in which it is omitted (Hock and Holmgren, 1996; Pellicciotti et al., 2009, Yang et al., 2021). Subsurface energy fluxes have been found to represent 1-2% of the total heat flux on glacier surfaces (Giesen et al., 2008; Yang et al., 2011; Zhang et a., 2017).

***Line 188: It rained at the summit of Greenland Ice Sheet this year, so probably better to rephrase this sentence.***
**[Author response]**
Part of the value we see in this study stems from the fact that the St. Elias Icefields are vastly different than Greenland, so our focus here is on observations specifically in the St. Elias.

***Line 195: Why assume $E_N$=0, when it will most certainly not be, either seasonally or annually?***
[Author response]
We assume $E_N$=0 because our aim here is simply to get an estimate of whether near-surface inversions are possible at our study site. At a site high in the accumulation zone with relatively limited melt, $E_N$=0 is an adequate approximation for this purpose. The implication of this assumption is that our modeled surface temperature may be underestimated in the summer and overestimated in the winter, meaning that wintertime inversions may be even more pronounced than our model results suggest.

***Line 200: Provide range of values for atmospheric emissivity.***
[Author response]
Updated text to read: We calculate downward longwave radiation as follows, using 2 m air temperature ($T_a$) from Divide and atmospheric emissivity ($\varepsilon_a$) derived from the ERA5 reanalysis longwave radiation product. We use only the derived emissivity from the ERA5 product, rather than the total downward radiation in order to use measured values (in situ 2 m air temperature) where possible. ERA5 outputs have a spatial resolution of 31 km; data span 2002-2019 every six hours (Hersbach et al., 2020). Atmospheric emissivity increases with increasing surface vapor pressure (Staley and Jurica, 1971). Our atmospheric emissivity values ranged from ~0.48 to 1. Atmospheric emissivity measured over the Sierra Nevada (Spain) from 2005-2011 ranged from ~0.4-1 (Herrero and Polo, 2012).

***Line 200: ERA 5 Land produces a downwelling longwave variable. Why wasn't this incorporated into the analysis?***
[Author response]
We chose not to incorporate the ERA5 downwelling longwave variable because we wanted to use in situ data wherever possible in the analysis.

***Equation 3: Provide more information about how this equation was derived. And why use a literature value for albedo? There is considerable variation, spatially and temporally, in albedo. Why not use the coincident MODIS albedo?***
[Author response]
Updated text to read: $E_L\uparrow$ is the energy emitted by the earth's surface and can be described by:

$$E_L\uparrow \ = \ c\sigma T_S{}^4$$

where $\epsilon$ is surface emissivity, $\sigma$ is the Stefan-Boltzmann constant, and $T_S$ is surface temperature (Cuffey and Paterson, 2010). Expressing $E_L\uparrow$ in terms of its components…

We used the mean measured albedo value from Divide in the interest of incorporating as much in situ data as possible into our analyses. Prior work in the St. Elias (Williamson et al., 2016) has demonstrated issues with MODIS albedo values arising from confusion between snow and cloud cover. We therefore avoided using the MODIS albedo product to eliminate this unnecessary source of uncertainty.

***Line 203: MODIS provides emissivity values. What are these for the given days sampled in this study? What are the seasonal ranges of snow emissivity?***
[Author response]
The emissivity values for the days sampled in this study range from 0.930 to 0.988. In spring (MAM) they range from 0.944 to 0.988. In summer (JJA) they range from 0.930 to 0.986. In fall (SON) they range from 0.942 to 0.988. In winter (DJF) they range from 0.942 to 0.986. Boxplots by season of the MODIS

emissivity values from bands 29, 31 and 32 are shown below. The range of emissivity values is similar in all seasons, so we consider distinguishing by season unnecessary for our simple model. The distribution of emissivity values is also skewed toward higher values, so we consider the 0.95 value from Hori et al. (2006) a reasonable choice for our lower emissivity bound.

[Figure]

***Line 208: Differences between median values? I am unsure what "Median differences" is.***
[Author response]
We first calculated the difference for each paired AWS measurement and MODIS LST and then took the median of these values.

Statement to be added to methods sections (around lines 145-150): We calculate the difference between each paired AWS measurement and MODIS LST to produce a timeseries of the offset between the two. We report the median values of the resultant timeseries for each season below.

Updated text to read: In comparing MODIS LSTs with AWS temperatures at Divide and Eclipse, we find the MODIS LST offset to be greatest during the fall and winter (Table 3). We report a warmer surface as a positive difference and a colder surface as a negative difference. The difference between AWS temperatures and MODIS LSTs at Divide is larger in the fall (Mdn = -4:43°C) and winter (Mdn = -8:40°C) than in the spring...

***Line 209: Which is warmer, surface or air? Not clear.***
[Author response]
See edited statement for line 208 above

***Line 210: Are these distributions normally distributed? There are tests to determine this.***
[Author response]
The distributions are not normal, which is why we used the Wilcoxon rank sum test instead of a two-sided T test.

***I gather that seasonal averages use all of the data from 2000-2020. Are air temperature and surface temperature changing at the same rate? Are inversions weakening over time? Are rates of temperature change similar between seasons? Is there a monotonic trend in emissivity? All of these things will influence your results.***
[Author response]

Yes, seasonal averages use all of the data from 2000-2020. There are no clear seasonal trends in air or surface temperature or their rates of change over this time period. Air and surface temperatures change at similar rates over this period. There is also no clear trend in the LST offset or monotonic trend in emissivity over this period. The most important consideration for our analysis of seasonal differences in the LST offset is whether the relationship between air and surface temperatures has exhibited different changes in each season over the period from 2000-2020, and we do not see this.

[Figure]

[Figure]

[Figure]

**Line 223: *Temperature has not been measured to the precision being reported.***
[Author response]
MODIS temperatures are measured to the 0.01 and temperatures from the Divide AWS are measured to the 0.001.

**Line 247: *R^2 =0.02 is statistically significant? How big was this data set?***
[Author response]
The dataset had 395 pairs. See the response to comment on Figure 7 below for more information. Despite a statistically significant $r^2$ between the temperature difference and wind speed, we find the correlation weak to the point of being negligible.

**Figure 5: *I am not sure what the point of this figure is? The two MODIS thermal bands will differentially absorb in the atmosphere, which is the basis for the split window LST algorithm. To work out the atm. emissivity, atm. column water vapour is required.***
[Author response]

The purpose of Figure 5 is to illustrate that the MODIS brightness temperatures show the same pattern of offset from in situ measurements (greater in fall and winter) as the LSTs do. Therefore, this seasonal offset doesn't come from the introduction of emissivity during the LST calculations.

***Figure 6: Are these data temporally matched? It must be sampled data because a daily average of 1000 w/m^2 is not feasible.***
[Author response]
Yes, the data are temporally matched. Wind speeds were taken for the dates/times that we had already calculated the MODIS LST offset from in situ measurements.

***Line 263: "averaging temperature" - means what?***
[Author response]
Updated text to read: The LST offset in ASTER data indicates that MODIS LSTs do not display an offset from AWS temperatures simply because they are mean temperatures over square kilometer grid cells rather than point measurements.

***Line 266: Air temperature scales over 100s of km, so not surprising.***
[Author response]
Perhaps unsurprising, but still worth noting, especially given the variability of the terrain in this area.

***Figure 7: I am skeptical about the magnitude of the p-values reported here. These should be checked.***
[Author response]
We checked the p-values and re-ran the correlations using different methods to transform the dataset to normal (log, exponent, boxcox). The boxcox method provided the best transform and the resulting correlations are shown below.

[Figure]

*Line 275-277: Why would you expect this?  Make sure all of the expectations in this section include enough background information for a reader to evaluate.*

[Author response]
We have changed the language below to be more direct instead of rambling through expected outcomes of hypothetical scenarios.

Updated text to read: We find similar seasonal distributions of offset from AWS temperatures in MODIS LSTs and MODIS brightness temperatures, suggesting that the preferential fall and winter offset is not introduced by the conversion from brightness temperature to surface temperature or the

emissivity values used in this conversion (Fig. 5). Moreover, Landsat brightness temperatures also show a pattern of greater offset from AWS temperatures in the fall and winter. The observed apparent cold bias in MODIS LSTs is therefore not unique to the MYD21 product or even the MODIS sensor. Unfortunately, due to the limited availability of ASTER data, too few images exist to examine any seasonal pattern.

*Figure 9: Earlier in the methods you said the time of MODIS capture was between 11AM and1:30PM. Why the different diurnal time range here? Same issue with Table 6.*
[Author response]
We use a wider diurnal time range in Figure 9 and Table 6 to illustrate why our MODIS LST offsets (calculated using individual time-matched data points) is smaller than previously studied LST offsets (calculated using daily averages). Figure 9 and Table 6 deal with inversions calculated from ERA 5 and Divide AWS data, not MODIS data.

*Line 289: "suggesting that emissivity values during these seasons may contribute to the offset" - how exactly?*
[Author response]
See response to comment on line 96 above

*If there is a trend in cloud cover change then both downwelling shortwave and longwave radiation will be altered over the course of the study period. This could add a substantial amount of error to the results. This needs to be analysed.*
[Author response]
All of our data are from days with cloud-free MODIS imagery, so the impact of a trend in cloud cover change would be a change in the distribution of our sampling dates. Distribution of our sampling dates is shown below by year and overall.

[Figure]

[Figure]

**Line 295: which changing surface conditions?**

[Author response]

Updated text to read: Emissivity values may be especially poorly known under winter conditions because of rapidly changing snow surface characteristics during and following frequent snowfall events, resulting in the seasonal difference in outcome of the LST algorithm as seen at Divide

**Line 314: "wintertime temperature inversion" level? Williamson et al. (2020) put inversion level at approximately 1200 masl. The two stations used here are 1000 to 2000 m above this level, and are not situated in valleys where cold air drains and collects.**

[Author response]

All uses of "inversion" in this study refer to near-surface inversions within 2 m of the snow surface, rather than large-scale inversions such as the ones mentioned in Williamson et al. (2020). We will insert a statement to clarify our usage of the term.

**Line 315: Why was a simple energy balance model used when radiosonde or re-analysis data can be used to determine inversion depth?**

[Author response]

See response to comment on Line 314 above. Radiosonde and re-analysis data have been used to study large-scale inversions, but we are focused on near-surface energy balance processes, which is why we use a simple energy balance model.

*Line 367: "Surface melt is primarily driven by high air temperature" - what is high? And melt is correlated with air temperature. There are many examples in the literature of melt rate being influenced by short and longwave radiation.*

[Author response]

Updated text to read: Surface melt correlates with air temperatures, largely because of increased longwave atmospheric radiation (an important source of energy for melt) with higher temperatures (Ohmura, 2001; Cuffey and Paterson, 2010). Higher air temperatures tend to occur under cloudy conditions when no MODIS imagery is available (Walsh and Chapman, 1998). MODIS LSTs may therefore be inadequate for examining temperature conditions associated with individual extreme melt events.

*Line 369-370: MODIS albedo correlates very well to glacier mass balance. There are many examples of this to be found in the literature. MODIS can't measure albedo under cloud cover. I am not sure the statement presented in the manuscript is correct.*

[Author response]

Although MODIS albedo may correlate well to glacier mass balance overall, it may not capture individual extreme melt events. See response to comment on Line 367 above.

*Figure 10: Corrected is the wrong word. LST and air temperature are not the same thing and should display offsets. These offsets are important for understanding the energy transfer between surface and atmosphere. If the goal is to produce air temperature fields originating from MODIS LST, then 'converted' instead of 'corrected' might be a better option. Further, there are many examples of methods to convert LST in the literature, most of which do not appear in the manuscript. The AWS data from 2020 is suspiciously cold.*

[Author response]

We agree that the use of "corrected" is misleading. We will change all usages of "corrected" to "converted". Our data for 2020 were incomplete at the time of acquisition, only running through June, so the warmest months were not included. For clarity, 2020 will be omitted.

Edited (Line 370): However, interannual trends in MODIS LSTs agree well with those in AWS temperatures ($r^2 = 0:23$ and $p < 0:05$; Fig. 10). Various methods have been used to convert MODIS LSTs to air temperatures, including advanced statistical and modeling frameworks (e.g. Hengl et al., 2012; Benali et al., 2012; Emamifar et al., 2013; Wenbin et al., 2013; Janatian et al., 2017; Zhang et al., 2016; Hooker et al., 2018; Zhang et al., 2018; Zhang et al., 2021) Here we apply a simple linear regression ($y = 3:35+0:49x$) to show that the difference between mean annual MODIS LSTs and AWS temperatures (mean error of 0.00±1.77 C) can be reconciled, effectively converting MODIS LSTs to air temperatures and enabling their use for both qualitative and quantitative applications related to glacier melt and mass balance on annual timescales.

*Line 376: Snow and ice melts when its temperature reaches 0°C not when the air temperature above it reaches 0°C. So the rationale here needs to be revisited.*

[Author response]

Snow and ice do indeed melt when their temperature reaches 0°C; however, the rate of snow and ice melt does also exhibit a close relationship with air temperatures (Ohmura, 2001). Numerous studies have used this relationship to reconstruct air temperatures using the presence and thickness of melt layers observed in the snowpack or ice core (e.g. Abram et al., 2013; Alley and Anandakrishnan, 1995; Das and Alley, 2008; Fisher et al., 1995; Herron et al., 1981; Winski et al., 2018).

**References:**

Abram, Nerilie J., Robert Mulvaney, Eric W. Wolff, Jack Triest, Sepp Kipfstuhl, Luke D. Trusel, Françoise Vimeux, Louise Fleet, and Carol Arrowsmith. 2013. "Acceleration of Snow Melt in an Antarctic Peninsula Ice Core during the Twentieth Century." *Nature Geoscience 2013 6:5* 6 (5): 404–11. https://doi.org/10.1038/NGEO1787.

Adolph, Alden C., Mary R. Albert, and Dorothy K. Hall. 2018. "Near-Surface Temperature Inversion during Summer at Summit, Greenland, and Its Relation to MODIS-Derived Surface Temperatures." *The Cryosphere* 12: 907–20. https://doi.org/10.5194/tc-12-907-2018.

Alley, R. B., and S. Anandakrishnan. 1995. "Variations in Melt-Layer Frequency in the GISP2 Ice Core: Implications for Holocene Summer Temperatures in Central Greenland." *Annals of Glaciology* 21: 64–70. https://doi.org/10.3189/S0260305500015615.

Benali, A., A. C. Carvalho, J. P. Nunes, N. Carvalhais, and A. Santos. 2012. "Estimating Air Surface Temperature in Portugal Using MODIS LST Data." *Remote Sensing of Environment* 124 (September): 108–21. https://doi.org/10.1016/J.RSE.2012.04.024.

Brock, Ben W., and Neil S. Arnold. 2000. "A Spreadsheet-Based (Microsoft Excel) Point Surface Energy Balance Model for Glacier and Snow Melt Studies." *Earth Surface Processes and Landforms* 25 (6): 649–58. https://doi.org/10.1002/1096-9837(200006)25:6<649::AID-ESP97>3.0.CO;2-U.

Budyko, M. I. 1969. "The Effect of Solar Radiation Variations on the Climate of the Earth." *Tellus* 21 (5): 611–19. https://doi.org/10.1111/J.2153-3490.1969.TB00466.X.

Chen, B., W. C. Chao, and X. Liu. 2003. "Enhanced Climatic Warming in the Tibetan Plateau Due to Doubling CO2: A Model Study." *Climate Dynamics* 20 (4): 401–13. https://doi.org/10.1007/S00382-002-0282-4.

Cuffey, K. M., and W. S. B. Paterson. 2010. *The Physics of Glaciers*. 4th ed. Elsevier.

Das, Sarah B., and Richard B. Alley. 2008. "Rise in Frequency of Surface Melting at Siple Dome through the Holocene: Evidence for Increasing Marine Influence on the Climate of West Antarctica." *Journal of Geophysical Research: Atmospheres* 113 (D2): 2112. https://doi.org/10.1029/2007JD008790.

Diaz, Henry F., Raymond S. Bradley, and Liang Ning. 2018. "Climatic Changes in Mountain Regions of the American Cordillera and the Tropics: Historical Changes and Future Outlook." *Https://Doi.Org/10.1657/1938-4246-46.4.735* 46 (4): 735–43. https://doi.org/10.1657/1938-4246-46.4.735.

Emamifar, Saeed, Ali Rahimikhoob, and Ali Akbar Noroozi. 2013. "Daily Mean Air Temperature Estimation from MODIS Land Surface Temperature Products Based on M5 Model Tree." *International Journal of Climatology* 33 (15): 3174–81. https://doi.org/10.1002/JOC.3655.

Farinotti, Daniel, Matthias Huss, Johannes J. Fürst, Johannes Landmann, Horst Machguth, Fabien Maussion, and Ankur Pandit. 2019. "A Consensus Estimate for the Ice Thickness Distribution

of All Glaciers on Earth." *Nature Geoscience* 12 (3): 168–73.
https://doi.org/10.1038/s41561-019-0300-3.

Favier, Vincent, Patrick Wagnon, Jean Philippe Chazarin, Luis Maisincho, and Anne Coudrain. 2004.
"One-Year Measurements of Surface Heat Budget on the Ablation Zone of Antizana Glacier 15,
Ecuadorian Andes." *Journal of Geophysical Research: Atmospheres* 109 (D18): 18105.
https://doi.org/10.1029/2003JD004359.

Fisher, D. A., R. M. Koerner, and N. Reeh. 2016. "Holocene Climatic Records from Agassiz Ice Cap,
Ellesmere Island, NWT, Canada:"
*Http://Dx.Doi.Org.Wv-o-Ursus-Proxy02.Ursus.Maine.Edu/10.1177/095968369500500103* 5
(1): 19–24. https://doi.org/10.1177/095968369500500103.

Giesen, R. H., M. R. van den Broeke, J. Oerlemans, and L. M. Andreassen. 2008. "Surface Energy
Balance in the Ablation Zone of Midtdalsbreen, a Glacier in Southern Norway: Interannual
Variability and the Effect of Clouds." *Journal of Geophysical Research: Atmospheres* 113
(D21). https://doi.org/10.1029/2008JD010390.

Giorgi, Filippo, James W. Hurrell, Maria Rosaria Marinucci, Martin Beniston, Filippo Giorgi, James W.
Hurrell, Maria Rosaria Marinucci, and Martin Beniston. 1997. "Elevation Dependency of the
Surface Climate Change Signal: A Model Study." *Journal of Climate* 10 (2): 288–96.
https://doi.org/10.1175/1520-0442(1997)010.

Hengl, Tomislav, Gerard B. M. Heuvelink, Melita Perčec Tadić, and Edzer J. Pebesma. 2012.
"Spatio-Temporal Prediction of Daily Temperatures Using Time-Series of MODIS LST
Images." *Theoretical and Applied Climatology* 107 (1–2): 265–77.
https://doi.org/10.1007/S00704-011-0464-2/FIGURES/8.

Herrero, J., and M. J. Polo. 2012. "Hydrology and Earth System Sciences Parameterization of
Atmospheric Longwave Emissivity in a Mountainous Site for All Sky Conditions." *Hydrol.
Earth Syst. Sci* 16: 3139–47. https://doi.org/10.5194/hess-16-3139-2012.

Herron, Michael M., Susan L. Herron, and Chester C. Langway. 1981. "Climatic Signal of Ice Melt
Features in Southern Greenland." *Nature 1981 293:5831* 293 (5831): 389–91.
https://doi.org/10.1038/293389a0.

Hersbach, Hans, Bill Bell, Paul Berrisford, Shoji Hirahara, András Horányi, Joaquín Muñoz-Sabater,
Julien Nicolas, et al. 2020. "The ERA5 Global Reanalysis." *Quarterly Journal of the Royal
Meteorological Society* 146 (730): 1999–2049. https://doi.org/10.1002/QJ.3803.

Hock, Regine, and Björn Holmgren. 1996. "Some Aspects of Energy Balance and Ablation of
Storglaciären, Northern Sweden." *Geografiska Annaler: Series A, Physical Geography* 78
(2–3): 121–31. https://doi.org/10.1080/04353676.1996.11880458.

Hock, Regine, and Christian Noetzli. 1997. "Areal Melt and Discharge Modelling of Storglaciären,
Sweden." *Annals of Glaciology* 24: 211–16. https://doi.org/10.3189/S0260305500012192.

Hooker, Josh, Gregory Duveiller, and Alessandro Cescatti. 2018. "A Global Dataset of Air Temperature Derived from Satellite Remote Sensing and Weather Stations." *Scientific Data 2018 5:1* 5 (1): 1–11. https://doi.org/10.1038/sdata.2018.246.

Hori, Masahiro, Teruo Aoki, Tomonori Tanikawa, Hiroki Motoyoshi, Akihiro Hachikubo, Konosuke Sugiura, Teppei J. Yasunari, et al. 2006. "In-Situ Measured Spectral Directional Emissivity of Snow and Ice in the 814 Mm Atmospheric Window." *Remote Sensing of Environment* 100 (4): 486–502. https://doi.org/10.1016/j.rse.2005.11.001.

Hudson, Stephen R., and Richard E. Brandt. 2005. "A Look at the Surface-Based Temperature Inversion on the Antarctic Plateau." *Journal of Climate* 18 (11): 1673–96. https://doi.org/10.1175/JCLI3360.1.

Hugonnet, Romain, Robert McNabb, Etienne Berthier, Brian Menounos, Christopher Nuth, Luc Girod, Daniel Farinotti, et al. 2021. "Accelerated Global Glacier Mass Loss in the Early Twenty-First Century." *Nature 2021 592:7856* 592 (7856): 726–31. https://doi.org/10.1038/s41586-021-03436-z.

Hulley, Glynn, Nabin Malakar, and Robert Freepartner. 2016. "Moderate Resolution Imaging Spectroradiometer (MODIS) Land Surface Temperature and Emissivity Product (MxD21) Algorithm Theoretical Basis Document," 102.

Janatian, Nasime, Morteza Sadeghi, Seyed Hossein Sanaeinejad, Elham Bakhshian, Ali Farid, Seyed Majid Hasheminia, and Sadegh Ghazanfari. 2017. "A Statistical Framework for Estimating Air Temperature Using MODIS Land Surface Temperature Data." *International Journal of Climatology* 37 (3): 1181–94. https://doi.org/10.1002/JOC.4766.

Koenig, Lora S., and Dorothy K. Hall. 2010. "Comparison of Satellite, Thermochron and Air Temperatures at Summit, Greenland, during the Winter of 2008/09." *Journal of Glaciology* 56 (198): 735–41. https://doi.org/10.3189/002214310793146269.

Li, Hua, Ruibo Li, Yikun Yang, Biao Cao, Zunjian Bian, Tian Hu, Yongming Du, Lin Sun, and Qinhuo Liu. 2020. "Temperature-Based and Radiance-Based Validation of the Collection 6 MYD11 and MYD21 Land Surface Temperature Products Over Barren Surfaces in Northwestern China." *IEEE Transactions on Geoscience and Remote Sensing*, 1–14. https://doi.org/10.1109/TGRS.2020.2998945.

Lian, M. S., and Robert D. Cess. 1977. "Energy Balance Climate Models: A Reappraisal of Ice-Albedo Feedback." *Article in Journal of the Atmospheric Sciences*. https://doi.org/10.1175/1520-0469(1977)034<1058:EBCMAR>2.0.CO;2.

McConnell, Erin A. 2019. "Mechanisms of Ice Core Stable Isotope Variability in the Upper Kaskawulsh-Donjek Region, St. Elias Mountains, Yukon, Canada." The University of Maine. https://digitalcommons.library.umaine.edu/etd/3069.

Ohmura, Atsumu. 2012. "Enhanced Temperature Variability in High-Altitude Climate Change." *Theoretical and Applied Climatology 2012 110:4* 110 (4): 499–508. https://doi.org/10.1007/S00704-012-0687-X.

Pellicciotti, F., M. Carenzo, J. Helbing, S. Rimkus, and P. Burlando. 2009. "On the Role of Subsurface Heat Conduction in Glacier Energy-Balance Modelling." *Annals of Glaciology* 50 (50): 16–24. https://doi.org/10.3189/172756409787769555.

Pepin, N., R. S. Bradley, H. F. Diaz, M. Baraer, E. B. Caceres, N. Forsythe, H. Fowler, et al. 2015. "Elevation-Dependent Warming in Mountain Regions of the World." *Nature Climate Change* 5 (5): 424–30. https://doi.org/10.1038/nclimate2563.

Rangwala, Imtiaz, and James R. Miller. 2012. "Climate Change in Mountains: A Review of Elevation-Dependent Warming and Its Possible Causes." *Climatic Change* 114 (3): 527–47. https://doi.org/10.1007/s10584-012-0419-3.

Serreze, Mark C., and Roger G. Barry. 2011. "Processes and Impacts of Arctic Amplification: A Research Synthesis." *Global and Planetary Change* 77 (1): 85–96. https://doi.org/10.1016/j.gloplacha.2011.03.004.

Shuman, Christopher A., Dorothy K. Hall, Nicolo E. DiGirolamo, Thomas K. Mefford, and Michael J. Schnaubelt. 2014. "Comparison of Near-Surface Air Temperatures and MODIS Ice-Surface Temperatures at Summit, Greenland (200813)." *Journal of Applied Meteorology and Climatology* 53 (9): 2171–80. https://doi.org/10.1175/JAMC-D-14-0023.1.

Staley, D. O., and G. M. Jurica. 1972. "Effective Atmospheric Emissivity under Clear Skies." *Journal of Applied Meteorology and Climatology* 11 (2): 349–56.

Walsh, John E., and William L. Chapman. 1998. "Arctic CloudRadiationTemperature Associations in Observational Data and Atmospheric Reanalyses." *Journal of Climate* 11 (11): 3030–45. https://doi.org/10.1175/1520-0442(1998)011<3030:ACRTAI>2.0.CO;2.

Williamson, Scott N., Luke Copland, and David S. Hik. 2016. "The Accuracy of Satellite-Derived Albedo for Northern Alpine and Glaciated Land Covers." *Polar Science*, ISAR-4/ICARPIII, Science Symposium of ASSW2015, 10 (3): 262–69. https://doi.org/10.1016/j.polar.2016.06.006.

Williamson, Scott N., David S. Hik, John A. Gamon, Alexander H. Jarosch, Faron S. Anslow, Garry K. C. Clarke, and T. Scott Rupp. 2017. "Spring and Summer Monthly MODIS LST Is Inherently Biased Compared to Air Temperature in Snow Covered Sub-Arctic Mountains." *Remote Sensing of Environment* 189 (February): 14–24. https://doi.org/10.1016/J.RSE.2016.11.009.

Williamson, Scott N., David S. Hik, John A. Gamon, Jeffrey L. Kavanaugh, and Saewan Koh. 2013. "Evaluating Cloud Contamination in Clear-Sky MODIS Terra Daytime Land Surface Temperatures Using Ground-Based Meteorology Station Observations." *Journal of Climate* 26 (5): 1551–60. https://doi.org/10.1175/JCLI-D-12-00250.1.

Williamson, Scott N., Christian Zdanowicz, Faron S. Anslow, Garry K. C. Clarke, Luke Copland, Ryan K. Danby, Gwenn E. Flowers, Gerald Holdsworth, Alexander H. Jarosch, and David S. Hik. 2020. "Evidence for Elevation-Dependent Warming in the St. Elias Mountains, Yukon, Canada." *Journal of Climate*, February. https://doi.org/10.1175/JCLI-D-19-0405.1.

Winski, Dominic, Erich Osterberg, Karl Kreutz, Cameron Wake, David Ferris, Seth Campbell, Mark Baum, et al. 2018. "A 400-Year Ice Core Melt Layer Record of Summertime Warming in the Alaska Range." *Journal of Geophysical Research: Atmospheres* 123 (7): 3594–3611. https://doi.org/10.1002/2017JD027539.

Winton, Michael. 2006. "Amplified Arctic Climate Change: What Does Surface Albedo Feedback Have to Do with It?" *Geophysical Research Letters* 33 (3). https://doi.org/10.1029/2005GL025244.

Yang, Min, Zhongqin Li, Muhammad Naveed Anjum, Xin Zhang, Yayu Gao, and Chunhai Xu. 2021. "On the Importance of Subsurface Heat Flux for Estimating the Mass Balance of Alpine Glaciers." *Global and Planetary Change* 207 (December): 103651. https://doi.org/10.1016/J.GLOPLACHA.2021.103651.

Yang, Wei, Xiaofeng Guo, Tandong Yao, Kun Yang, Long Zhao, Shenghai Li, and Meilin Zhu. 2011. "Summertime Surface Energy Budget and Ablation Modeling in the Ablation Zone of a Maritime Tibetan Glacier." *Journal of Geophysical Research: Atmospheres* 116 (D14). https://doi.org/10.1029/2010JD015183.

Yao, Rui, Lunche Wang, Shaoqiang Wang, Lizhe Wang, Jing Wei, Junli Li, and Deqing Yu. 2020. "A Detailed Comparison of MYD11 and MYD21 Land Surface Temperature Products in Mainland China." *International Journal of Digital Earth* 0 (0): 1–17. https://doi.org/10.1080/17538947.2019.1711211.

You, Qinglong, Ziyi Cai, Nick Pepin, Deliang Chen, Bodo Ahrens, Zhihong Jiang, Fangying Wu, et al. 2021. "Warming Amplification over the Arctic Pole and Third Pole: Trends, Mechanisms and Consequences." *Earth-Science Reviews* 217 (June): 103625. https://doi.org/10.1016/J.EARSCIREV.2021.103625.

Zemp, M., M. Huss, E. Thibert, N. Eckert, R. McNabb, J. Huber, M. Barandun, et al. 2019. "Global Glacier Mass Changes and Their Contributions to Sea-Level Rise from 1961 to 2016." *Nature* 568 (7752): 382–86. https://doi.org/10.1038/s41586-019-1071-0.

Zhang, Guoshuai, Shichang Kang, Koji Fujita, Eva Huintjes, Jianqing Xu, Takeshi Yamazaki, Shigenori Haginoya, et al. 2013. "Energy and Mass Balance of Zhadang Glacier Surface, Central Tibetan Plateau." *Journal of Glaciology* 59 (213): 137–48. https://doi.org/10.3189/2013JoG12J152.

Zhang, Hongbo, Fan Zhang, Ming Ye, Tao Che, and Guoqing Zhang. 2016. "Estimating Daily Air Temperatures over the Tibetan Plateau by Dynamically Integrating MODIS LST Data." *Journal of Geophysical Research: Atmospheres* 121 (19): 11,425-11,441. https://doi.org/10.1002/2016JD025154.

Zhang, Hongbo, Fan Zhang, Guoqing Zhang, Yaoming Ma, Kun Yang, and Ming Ye. 2018. "Daily Air Temperature Estimation on Glacier Surfaces in the Tibetan Plateau Using MODIS LST Data." *Journal of Glaciology* 64 (243): 132–47. https://doi.org/10.1017/JOG.2018.6.

Zhang, Wenjie, Baiping Zhang, Wenbin Zhu, Xiaolu Tang, Fujie Li, Xisheng Liu, and Qiang Yu. 2021. "Comprehensive Assessment of MODIS-Derived near-Surface Air Temperature Using Wide Elevation-Spanned Measurements in China." *Science of The Total Environment* 800 (December): 149535. https://doi.org/10.1016/J.SCITOTENV.2021.149535.

---

## Author Response (AR2)

Thank you for your comments, which we have addressed line by line below. Comments are indicated by boldface and italicized text; our responses in normal text and preceded by [Author response].

***L5: Are the "available weather station measurements" referring to air or skin temperatures? Or perhaps both? I think it would be useful to specify that here.***

[Author response]
Weather station measurements have been specified as air temperatures.

***L108: I believe this is an incomplete sentence, or needs to be rewritten for clarity.***

[Author response]
The sentence has been completed- whoops!

***Table 1: The MYD21 product does have a layer that reports LST error that could be added to this table (https://lpdaac.usgs.gov/products/myd21v061/). This would likely help to contribute to the arguments that the offsets that are calculated between MODIS LST and air temperature are not simply due to measurement error--if the difference is larger than the combined error of both datasets (MODIS LST and air temperature), which presumably it is.***

[Author response]
We chose not to alter Table 1, but instead to insert a statement that all measurement errors on the MODIS LSTs are <1°C.

***L154: It would be useful to add what MYD21 version you are using. v006? V061?***

[Author response]
"v006" has been added.

***L173: Were MODIS LST values ever above 0°C? If so, were they left as is?***

[Author response]
All MODIS LST values were left as is. A statement to this effect has been added.

***L256: "MODIS LSTs and melt" – I think the ability to use annual MODIS LST averages to approximate annual average air temperature extends beyond using it to determine potential correlations with melt (especially since the comparison with melt events isn't explicitly done in this work). I think focusing more generally on how this linear regression could be used to approximate air temperature would broaden the applicability. Then I do think the focus on melt in the discussion section is reasonable and gives a specific instance where this might be useful. However, I do wonder if MODIS LST (unconverted) is more correlated with melt events than the approximated air temperature. I understand that the cited literature focuses on links between air temperature and melt, but it seems a surface temperature is in fact more directly linked to potential melt events. I am not sure if data exist to test this, but it could be mentioned in the discussion.***

[Author response]
The subsection heading has been changed to "Approximating air temperatures from MODIS LSTs" and broader context has been added before diving into surface melt.

***L262-266: This is simply an issue of wording, but I think calling the approximated air temperature***

*"LST_{converted}" is a bit confusing. This is an approximation of air temperature based on LST. So calling it "T_{air, approx.}" or something might be more clear.*

[Author response]
$T_{converted}$ has been changes to $T_{air, approx.}$.

*L379: Typo, should read "fall and winter"*

[Author response]
Now reads "fall and winter"

*L453: enabling --> enables*

[Author response]
"and enabling" has been changed to ", enabling"